# EVEREST 🏔: Efficient Masked Video Autoencoder by Removing Redundant Spatiotemporal Tokens

## Abstract

Masked video autoencoder approaches have demonstrated their potential by significantly outperforming previous self-supervised learning methods in video representation learning. However, they require an excessive amount of computations and memory while predicting uninformative tokens/frames due to random masking strategies, requiring excessive computing power for training. (e.g., over 16 nodes with 128 NVIDIA A100 GPUs (Feichtenhofer et al., 2022)). To resolve this issue, we exploit the unequal information density among the patches in videos and propose *Efficient Masked Video AutoEncoder by Removing REdundant Spatiotemporal Tokens (EVEREST)*, a new token selection method for video representation learning that finds tokens containing rich motion features and drops uninformative ones during both pre-training and fine-tuning. We further present an information-intensive frame selection strategy that allows the model to focus on informative and causal frames with minimal redundancy. Our method significantly reduces computation and memory requirements of Masked video autoencoder, enabling the pre-training and fine-tuning on a single machine with 8 GPUs while achieving comparable performance to computation- and memory-heavy baselines on multiple benchmarks and on the uncurated Ego4D dataset. We hope that our work contributes to reducing the barrier to further research on video understanding.

## 1 Introduction

Massive video data floods the web and media daily with the rapid growth of portable devices equipped with cameras, such as AR glasses, smartphones, UAVs, and robots. However, direct utilization of user-generated video data for solving target task problems is nontrivial, as annotating them is time-consuming and expensive. One potential approach to tackle this problem is to learn generic representations from unlabeled video streams that can transfer to diverse downstream visual tasks. Video Representation Learning (VRL) (Fernando et al., 2017; Piergiovanni et al., 2020; Qian et al., 2021; Pan et al., 2021) methods allow learning spatial and temporal features from input video frames in a self-supervised manner without any human annotations. A caveat to such pre-training for video tasks is that, unlike image-based tasks with static information of objects in instances, video-based tasks involve temporal causality; that is, successive frames are strongly correlated in their semantics.

Recently, Masked Video Autoencoder (MVA) (Tong et al., 2022; Feichtenhofer et al., 2022), which learns to reconstruct randomly masked spatiotemporal regions in video clips, has shown impressive performance on various video-based problems. Yet, critical challenges remain in efficiently exploiting the spatiotemporal information in real-world videos: (1) Tokens (a pair of two temporally successive patches in the same space) in videos are not equally valuable to reconstruct, as the amount of their information not only depends on spatial importance but also on temporal redundancy. (2) Learning representations from videos is infeasible without a huge computing budget. MVA approaches (Feichtenhofer et al., 2022; Wang et al., 2023b;c) that reconstruct the *whole* video require excessively large amounts of computations, making it impractical without access to a substantial GPU cluster. For example, VideoMAE (Tong et al., 2022) takes about 27 hours to pre-train for 800 epochs with ViT-B using **64 NVIDIA V100 (32GB) GPUs**, and ST-MAE (Feichtenhofer et al., 2022) takes about 35.8 hours to pre-train for 800 epochs with ViT-L using **128 NVIDIA A100 (80GB) GPUs**.

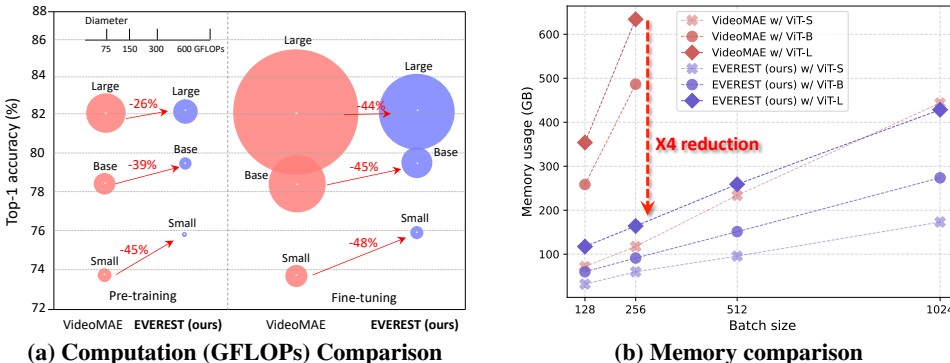

(a) Computation (GFLOPs) Comparison   (b) Memory comparison

Figure 1: **Efficiency of our EVEREST against VideoMAE on K400 dataset. (a) GFLOPs** for pre-training and fine-tuning. The bubble size is proportional to the GFLOPs of the model. **(b) Memory consumption** using one node equipped with $8\times$ A100 (80GB). VideoMAE with ViT-B and -L fails to deploy the model due to **out-of-memory** if the batch sizes are 512 or larger. For a ViT-L backbone with a batch size of 256, our method achieves about $4\times$ less memory consumption than VideoMAE. Please see Tables 1 and 3 for detailed results.

To overcome such limited feasibility of video representation learning, we propose an *Efficient Masked Video AutoEncoder which removes REdundant Spatiotemporal Tokens (EVEREST)*, that is a highly efficient training algorithm with token selection for alleviating the extremely high resource requirements for VRL. Unlike prior methods (Tong et al., 2022; Feichtenhofer et al., 2022; Wang et al., 2023a) reconstructing *all* patches, we aim to select a subset of *informative* visual tokens to learn based on the distance disparity across temporally adjacent tokens in the embedding space. Our *redundancy-robust token selection* approach successfully detects meaningful changes in the state/object of incoming videos in an online manner, discarding less meaningful tokens containing redundant information or meaningless backgrounds, without resorting to dense motion features from the incoming video, such as HOG or optical flows (Sun et al., 2023; Wang et al., 2023c).

This allows us to significantly reduce computational cost and memory usage while maintaining the quality of a representation model by back-propagating to only a few selected tokens that retain rich spatiotemporal information with minimal redundancy. As shown in Figues 1, our EVEREST **saves computation costs by $26 \sim 45\%$ in pre-training and $44 \sim 48\%$** in fine-tuning across varying ViT scales while achieving competitive performance against strong VideoMAE baselines. We also achieve an impressive amount of memory reduction with all ViT backbones, which becomes more effective with larger models. For example, on a ViT-Large with a batch size of 256, our method achieves about $4\times$ **smaller memory consumption** than VideoMAE and **enables single-node training with larger batch sizes** using a large backbone, whereas existing VRL methods require immense memory occupancy and fail to train in the same setup as they go *out-of-memory*.

In addition, most current VRL (Arnab et al., 2021; Bertasius et al., 2021; Feichtenhofer et al., 2022; Tong et al., 2022) methods uniformly load frames at regular time intervals from each mini-batch clip. However, this strategy does not consider a temporal imbalance in information and noise that matters to real-world uncurated videos. Our EVEREST further performs *information-intensive frame selection*, which is carried out online through *probabilistic sampling* proportional to the ratio of redundancy-robust tokens in each frame and does not need any additional computational or parametric learning steps. Consequently, we can capture abundant spatiotemporal knowledge from highly sparse yet informative spatiotemporal regions in videos.

We extensively validate our proposed method on multiple benchmark datasets, including UCF101, HMDB51, K400, Something-Something V2, and Ego4D, and our EVEREST shows remarkable efficiency in terms of memory occupancy, computational cost, and training time compared to strong counterparts, achieving competitive performance.

Our contributions are as follows:

- We propose *redundancy-robust token selection*, an *efficient* VRL method that promptly selects the most informative tokens based on the states' change and discards the redundant ones in an online manner, to avoid wasteful training on uninformative spatiotemporal regions of videos.

- We further propose *information-intensive frame selection*, a strategy to select informative video frames from incoming videos, which allows the model to efficiently learn robust and diverse temporal representations in real-world uncurated videos.

- Our EVEREST has great potential to lower the barrier for video-related research that requires enormous computing power and cost, as it shows comparable performance to existing methods while significantly reducing the amount of computations, memory usage, and training time.

## 2 RELATED WORK

**Masked video autoencoder** Inspired by self-supervised learning with Masked Image Modeling (He et al., 2022; Xie et al., 2021; Kakogeorgiou et al., 2022), several recent works on video representation learning (Wang et al., 2022b; Sun et al., 2023; Wang et al., 2023a;c) suggest spatiotemporal masking strategies given video streams. To capture spatial representation and temporal dynamics for unsupervised video streams, ST-MAE (Feichtenhofer et al., 2022) and VideoMAE (Tong et al., 2022) extend a masked image autoencoder to mask partial regions in arriving video clips via random and space-only random sampling, respectively. They find that spatiotemporal inductive bias in video clips helps a decoder predict input pixels in masked regions, allowing a higher masking ratio ($\sim 90\%$) than MIM ($\sim 60\%$ (Xie et al., 2021) and $\sim 75\%$ (He et al., 2022)) on image self-supervised learning. BEVT (Wang et al., 2022b) proposes to train image- and video-level masked autoencoders jointly by sharing weights of the encoder, formulated with Video Swin (Liu et al., 2022). They resort to random sampling given spatiotemporal inductive bias, which can be a good approximator with stochasticity during data-driven training. Nevertheless, selecting random tokens to reconstruct for Masked video autoencoder is inefficient since embedded tokens in video frames are not equally important, especially since the informativeness of each token is affected by the adjacent frames.

**Input selective training** As benchmark training datasets often have massive scales and contain a lot of redundant or less meaningful instances, various works (Fayyaz et al., 2022; Wu et al., 2019; Yoon et al., 2022) have discussed sampling important instances from the entire dataset or focusing on localized information in each frame (Meng et al., 2022; Fayyaz et al., 2022; Kakogeorgiou et al., 2022; Yin et al., 2022) for efficient image recognition. However, they have no means to capture a temporal correlation across adjacent frames in video tasks. A few works recently suggest *supervised input selection* techniques for video tokens or frames. Wang et al. (2022a) introduce a lightweight scorer network to select the most informative temporal and spatial token in incoming videos for supervised video classification tasks. Park et al. (2022) suggest the greedy K-center search that iteratively finds the most distant patches in the geometrical space from video clips. Gowda et al. (2021) train a single and global frame selector based on the ground truth for computing the importance score of single and paired frames. Zhi et al. (2021) performs a frame selection for unsupervised videos, but they have to extract whole frames from the training video dataset in advance to compute the change of cumulative distribution in video frames and features, consuming substantial pre-processing time and storage for saving the extracted frames (it takes **two days** to extract SSv2 into frames and occupies **433 GB**). Similarly, a few recently proposed MVA methods with adaptive token sampling require extracting dense motion information in advance or learnable parameters. MGM (Fan et al., 2023), MGMAE (Huang et al., 2023) and MME (Sun et al., 2023) generate motion-guided masking maps to reconstruct the informative tokens of the given videos, but they require motion vectors and optical flows, respectively.

However, **extracting all video frames and computing their motion information (e.g., HoG or optical flows) in advance is unrealistic** for online frame selection in videos. On the other hand, our EVEREST can perform rapidly without time- and memory-wasting steps before training.

## 3 VIDEO REPRESENTATION LEARNING VIA MASKED AUTOENCODERS

### 3.1 MASKED VIDEO AUTOENCODER

Learning to reconstruct intentionally corrupted data with masking is broadly utilized as means of representation learning in Natural Language Processing (Devlin et al., 2018; Song et al., 2019; Guu et al., 2020; Song et al., 2020) and has demonstrated its efficacy and power in broad research problems. In vision tasks, Masked Image Modeling (MIM) (He et al., 2022; Xie et al., 2021) aims to learn representations of the input images by solving the regression problem in which a model predicts RGB pixel values in randomly masked patch regions of images. The model divides an image into equally sized patches and then randomly chooses them to be masked based on a predetermined ratio.

Given unmasked patches, the encoder transforms them into feature vectors, and a decoder aims to reconstruct the original input images by predicting the RGB values of the missing image patches.

While the primary approach to VRL has been contrastive learning, the recent success of MIM has led to breakthroughs in effectively capturing spatiotemporal information from incoming video streams, which we call masked video autoencoder (Tong et al., 2022; Feichtenhofer et al., 2022). Let $\boldsymbol{v} \in \mathbb{R}^{2\tau \times C \times H \times W}$ be a short clip consisting of $2\tau$ successive frames from the entire video. A self-supervised model $f$ be formulated into an encoder-decoder framework $f(\cdot) = D(E(\cdot))$ aims to reconstruct partially masked frames in $\boldsymbol{v}$, guided by spatial and temporal relationships between tokens in adjacent frames. The encoder takes tokenized embedding vectors from a pair of successive frames using a 3D convolutional operation (Gupta et al., 2022; Ma et al., 2022; Piergiovanni et al., 2022; Qing et al., 2023; Wang et al., 2023b). Let $\boldsymbol{k}_i$ be an $i$-th spatiotemporal embedding vector for a pair of two frames $\boldsymbol{v}[2i : 2i + 1]$. Given $\boldsymbol{v}$, we reformulate the loss function as follows:

$$\ell(\boldsymbol{v}) = \sum_{i=0}^{\tau-1} \|D(E(\boldsymbol{m}_i \otimes \boldsymbol{k}_i; \mathcal{W}_E); \mathcal{W}_D) - (\mathbf{1} - \boldsymbol{m}_i) \otimes \boldsymbol{v}[2i : 2i + 1]\|_p, \tag{1}$$

$$\text{where } \boldsymbol{m} = G(J, \rho, \tau) \in \{0, 1\}^{\tau \times J},$$

where $G(\cdot)$ is a masking function depending on a specific policy, for example, random (or agnostic) (Tong et al., 2022) and space-only (or tube) (Feichtenhofer et al., 2022) masking techniques.

$J = \lfloor \frac{H}{s} \rfloor \cdot \lfloor \frac{W}{s} \rfloor$ denotes the number of patches per image and $\|\cdot\|_p$ denotes $p$-norm. $\mathcal{W}_E$ and $\mathcal{W}_D$ are a set of weights in the encoder and decoder, respectively. $\otimes$ is a dimensionality-preserving vector-matrix multiplication operation. A mask $\boldsymbol{m}_i \in \{0, 1\}^J$ is drawn by the binary distribution $B$ with a probability of $\rho$ without replacement, that is $|\boldsymbol{m}_i| = \lceil J \cdot \rho \rceil$. After self-supervised pre-training to minimize Equation 1, the encoder transfers the learned representations to various downstream tasks. Unlike the samples from the image dataset, which are permutation-invariant as they are independent of each other, consecutive frames from the video stream inherently have a strong correlation and redundancy. Thus, masked video autoencoder can enjoy spatiotemporal inductive bias from other adjacent frames in the input clip, achieving good reconstruction quality even with lessened hints (i.e., a proportion of unmasked tokens). Indeed, MVA allows a much higher masking ratio $1 - \rho$ per video against MIM methods. This property is advantageous because masked modeling with a higher masking ratio significantly reduces computations when training the encoder-decoder framework.

### 3.2 CHALLENGES IN MASKED VIDEO AUTOENCODER

MVA methods (Tong et al., 2022; Feichtenhofer et al., 2022; Wang et al., 2023a) basically capture meaningful representations from pre-training videos via random masking strategies for input tokens. These techniques are reasonable for curated and distributionally stable video datasets, yet, there is plenty of room for further development to make the model much more robust and computation-efficient. Here, we summarize two major **limitations** in the random sampling-based MVA:

**(1) Patchified image tokens from a video clip are *not equally* important.** At each iteration, MVA methods determine which tokens to mask according to specific random selection strategies (e.g., random, time-only, etc.). Yet, the relative amount of information in each token highly depends on the position of the informative objects and the correlation across patches within adjacent frames, which renders most of the tokens highly uninformative or redundant. These limitations lead to consuming massive training budgets in memory occupancy and suffering from slow convergence speed (e.g., training $3,200$ and $4,800$ epochs when using VideoMAE (Tong et al., 2022) on UCF101 and HMDB51, respectively). To address this information imbalance in visual tokens, several recent works have suggested sparsification/merging methods. EViT (Liang et al., 2022) is a supervised training method that fuses tokens by removing uninformative ones from the target task, therefore, inapplicable to video self-supervised learning. Token merging methods, including ToMe (Bolya et al., 2023), average multiple correlated/clustered tokens and let them have the same and marginalized values using average pooling. However, the masked video autoencoder encodes a few unmasked regions of video frames and aims to reconstruct raw frames. That is, token merging techniques are inappropriate for the decoding (i.e., reconstructing) phase of masked autoencoder by design. Furthermore, the encoding phase with token merging may also fail to reconstruct raw frames at a pixel-level, due to their marginalized token features. This technical design is well-performed in the action recognition classification problem but deteriorates the performance of the VRL method.

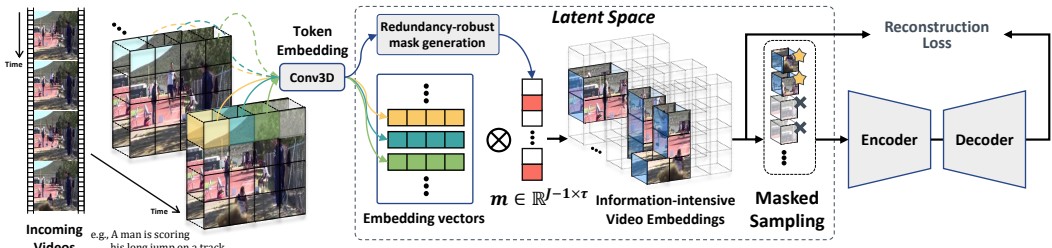

Figure 2: **Overview of our proposed approach**. Our redundancy-robust mask generator selects tokens with a large disparity with the paired ones in the previous time dimension, indicating that they include rich motion features. Then, the model focuses on learning spatiotemporal representation by reconstructing only sparsified videos containing abundant spatiotemporal information, which makes the VRL procedure surprisingly efficient.

**(2) MVAs draw frames at *uniform time intervals* from video streams to train on.** Real-world videos may include noisy and highly redundant frame sequences, often uninformative or even detrimental in representing temporal causal relationships and features of moving objects. However, the uniform sampling of video frames results in a waste of computational and memory resources in video understanding. To alleviate this problem, AdaFrame (Wu et al., 2019) trains the memory-augmented LSTM for gathering informative frames in given videos, leveraging supervised video labels to evaluate whether observing more frames is expected to produce more accurate predictions. SCSampler (Korbar et al., 2019) also requires the ground truth to jointly train the clip classifier and clip-level saliency model to obtain useful clips to the clip classifier. SMART (Gowda et al., 2021) trains Single-frame Selector and Global Selector for the frame selection, which require the ground truth for computing the importance score of single and paired frames.

These challenges further exacerbate the problem when applying masked video autoencoder on *uncurated* first-person view real-world videos, *e.g.,* Ego4D (Grauman et al., 2022), which **contains not only sparse motion information over space and time, but also suffers from a severe spatiotemporal redundancy.** Therefore, determining which frames and spatiotemporal tokens to recover is crucial for practical and efficient video representation learning.

## 4 EFFICIENT MASKED VIDEO AUTOENCODER BY REMOVING SPATIOTEMPORAL REDUNDANCY

### 4.1 REDUNDANCY-ROBUST MASK GENERATION FOR HIGHLY EFFICIENT MVA

Valuable spatiotemporal information in video streams mainly comes from active visual changes rather than from static backgrounds. Thus, we aim to learn self-supervision on videos from only a few crucial regions containing minimal spatiotemporal redundancy. Let $\boldsymbol{k}_i$ be a token embedding of a pair of adjacent $2i^{th}$ and $(2i+1)^{th}$ frames from an input video clip $\boldsymbol{v} \in \mathbb{R}^{2\tau \times C \times H \times W}$, where $0 \leq i < \tau$. $\boldsymbol{k}_{i+1,j}$ indicates the $j^{th}$ token embedding of $\boldsymbol{k}_{i+1}$ and we measure the importance $I_{i+1,j}$ of $\boldsymbol{k}_{i+1,j}$ by computing the *distance* from the token embedding at the same region in the previous time step, $\boldsymbol{k}_{i,j}$:

$$I_{i+1,j} = S\left(\boldsymbol{k}_{i+1,j}, \boldsymbol{k}_{i,j}\right), \text{ where } \boldsymbol{k}_i = \texttt{Conv3d}(\boldsymbol{v}[2i:2i+1]; \boldsymbol{w}), \qquad (2)$$

where $S(\cdot, \cdot)$ indicates a distance function, such as Euclidian, negative Cosine Similarity, and negative Centered Kernel Alignment (Cortes et al., 2012). Throughout this paper, we use the $\ell_2$ norm for $S$, which is simple yet empirically performs well, and we provide a discussion for the choice of the $S$ in Appendix B. Tokens with a large disparity from tokens in a previous time period are considered important, indicating that they contain unique knowledge in the video and may have more important information than other tokens. Given $\boldsymbol{v}$, the model determines the token embedding vectors with the highest importance ratio of $\rho_{pre}$ based on Equation 2, which we call *Redundancy-robust (ReRo) Masking Generation*. We can drastically reduce the computational cost during training by **only propagating these selected embedding vectors $\widetilde{\boldsymbol{k}}$** in a minibatch video clip at each iteration.

Next, we randomly sample a few token embeddings $\widetilde{\boldsymbol{k}}'$ from $\widetilde{\boldsymbol{k}}$ with the ratio of $\rho_{post}$ to forward them to the encoder. And the corresponding decoder predicts RGB pixels on all embedding vectors $\widetilde{\boldsymbol{k}}_i \in \widetilde{\boldsymbol{k}}$ based on the encoder outputs. We set a high masking ratio in general, $\rho_{pre} \cdot \rho_{post} = 0.1$, to follow

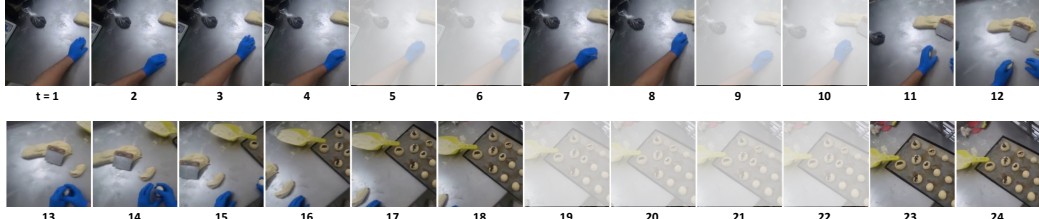

Figure 3: **Visualization of the proposed information-intensive frame selection on an uncurated dataset, Ego4D.** Unlike prior works that uniformly samples frames similar to each other, we adaptively sample the given video (24 frames) by probabilistic sampling the frames that have distinct spatiotemporal features (non-blurred frames).

the settings of our MVA baselines. The objective function of our EVEREST is formulated as follows:

$$arg \min_{\mathcal{W}, \boldsymbol{w}} \sum_{n=1}^{N} \sum_{i=0}^{\tau-1} \left\| f_{\mathcal{W}}((\widetilde{\boldsymbol{k}}_i')^n) - \boldsymbol{m}_i^n \otimes \boldsymbol{v}_n[2i : 2i+1] \right\|_F, \quad (3)$$

$$\text{where } \boldsymbol{m}^n = G\left([J \cdot \rho_{pre}], \rho_{post}, \tau\right) \in \{0, 1\}^{\tau \times [J \cdot \rho_{pre}]}.$$

We omit the positional embedding term for notational simplicity. Whereas earlier MVA works used a fixed masking ratio per video cues for self-supervised training, our approach allows a dynamic masking rate for each frame by design, based on the occupancy of essential tokens. As shown in Figues 7, each video frame contains a different amount of spatiotemporal information, and our dynamic masking strategy enables the model to focus on learning the valuable representation in a holistic view from incoming videos. We note that our EVEREST can be generalized well on both pre-training and fine-tuning phases with a negligible additional computational cost and training time (Please see Figues 1). We find that removing unimportant tokens is crucial for both phases, obtaining more meaningful representations and achieving superior performances. (Please see Table 16). The overall process of our redundancy-robust token selection is illustrated in Figues 2.

We emphasize that our ReRo mask generation is also significantly effective for motion-centric videos, e.g., Ego4D, since these *untrimmed real-world videos also contain a lot of temporally redundant visual information*. For example, in Figues 8, a worker focuses on a grass mower to operate it well, and a person makes cookie dough, where visual scenes include much meaningless visual information, like empty space on the table.

## 4.2 ON-THE-FLY INFORMATION-INTENSIVE FRAME SELECTION

As discussed in Section 3.2, real-world video streams may contain many redundant frames and also often include non-useful intermediate clips, such as temporally glancing at uninspiring walls, grounds, or skies, that interfere with estimating a causality of the user's or cameraman's attention. However, most video-based learning methods follow a simple strategy to sample frames from uniform time intervals at each iteration. This approach is reasonable for well-curated video benchmark datasets (Kay et al., 2017; Soomro et al., 2012) containing only information-dense frames with fixed viewpoints; however, it is unsuitable for uncurated real-world videos, which are more likely to have redundant and overlapped frames that the masked-video model can easily reconstruct.

To overcome the limitation, we propose to adaptively discard uninformative frames in the arrival video and build causal clips that represent the most crucial behaviors of the objects. We illustrate a simple overview of our *information-intensive frame selection* in Figues 4 and visualize the sampled results in Figues 3. We first select evenly spaced $[2\alpha \cdot \tau]$ frames as candidates, $\alpha$ times larger than the input clip size $\tau$, where $\alpha > 1$. We count the number of chosen tokens $c_i$ for the $2i^{th}$ and $(2i+1)^{th}$ frames based on the frequency of our *ReRo tokens*, described in Section 4.1. Then, the model iteratively trains on the input clip by drawing $\tau$ frames from $[\alpha \cdot \tau]$ candidates without replacement, with a probability of $\frac{c_i}{\sum_i c_i}$ that the paired

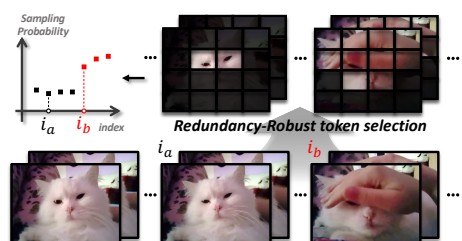

Figure 4: Illustration of **information-intensive frame selection.** We adaptively select frames based on the **ReRo** token frequency, which indicates significant spatiotemporal information compared to frames.

Table 1: **Comparison of fine-tuning Performance on K400.** For quick verification, VideoMAE and EVEREST are pre-trained for 200 epochs. PT and FT mean pre-training and fine-tuning, respectively. We use an input size of $16 \times 224^2$. Memory usage is measured with 256 batch size when pre-training. $\dagger$ is trained from random initialization. We refer Wightman (2019) for computing $\ddagger$.

| Method | Backbone | PT-Data | PT-GFLOPs | FT-GFLOPs | Memory usage (GB) | Top-1 Acc |
|---|---|---|---|---|---|---|
| MViT (Fan et al., 2021)$^\dagger$ | MViT-S | ✗ | - | 32.9 | - | 76.0 |
| MViT (Fan et al., 2021)$^\dagger$ | MViT-B | ✗ | - | 70.5 | - | 78.4 |
| ViViT FE (Arnab et al., 2021) | ViT-L | IN-21K | 119.0$^\ddagger$ | 3980.0 | N/A | 81.7 |
| K-centered (Park et al., 2022) | XViT | IN-1K | 67.4$^\ddagger$ | 425.0 | N/A | 73.1 |
| K-centered (Park et al., 2022) | Mformer | IN-1K | 67.4$^\ddagger$ | 369.5 | N/A | 74.9 |
| K-centered (Park et al., 2022) | TSformer | IN-1K | 67.4$^\ddagger$ | 590.0 | N/A | 78.0 |
| VideoMAE (Tong et al., 2022) | ViT-S | K400 | 11.6 | 57.0 | 117.4 | 73.5 |
| VideoMAE (Tong et al., 2022) | ViT-B | K400 | 35.5 | 180.5 | 486.4 | 78.4 |
| VideoMAE (Tong et al., 2022) | ViT-L | K400 | 83.1 | 597.2 | 634.1 | 82.0 |
| **EVEREST (ours)** | ViT-S | K400 | **6.3** (↓ 45.7%) | **29.1** (↓ 48.9%) | **59.9** (↓ 49.0%) | **75.9** |
| **EVEREST (ours)** | ViT-B | K400 | **21.5** (↓ 39.4%) | **98.1** (↓ 45.7%) | **91.2** (↓ 81.3%) | **79.2** |
| **EVEREST (ours)** | ViT-L | K400 | **60.8** (↓ 26.8%) | **330.0** (↓ 44.7%) | **164.1**(↓ 74.1%) | **82.1** |

Table 2: **Performance comparison with strong baselines on UCF101, HMDB51, and SSv2 datasets** without using the pre-training step on a large-scale dataset. Several results are drawn from Diba et al. (2021) and Tong et al. (2022). *SR50* indicates *SlowOnly-R50.*

| Method | Backbone | Extra data | T1 Acc (%) UCF101 | T1 Acc (%) HMDB51 | T1 Acc (%) SSv2 |
|---|---|---|---|---|---|
| VCOP (Xu et al., 2019) | R(2+1)D | UCF101 | 72.4 | 30.9 | N/A |
| CoCLR (Han et al., 2020) | S3D-G | UCF101 | 81.4 | 52.1 | N/A |
| Vi$^2$ CLR(Diba et al., 2021) | S3D | UCF101 | 82.8 | 52.9 | N/A |
| CoCLR (Han et al., 2020) | S3D-G | K400 | 87.9 | 54.6 | N/A |
| Vi$^2$ CLR(Diba et al., 2021) | S3D | K400 | 89.1 | 55.7 | N/A |
| RSPNet (Chen et al., 2021) | S3D-G | K400 | 93.7 | 64.7 | 55.0 |
| $\rho$SwAV$_{\rho=2}$ (Feich. et al., 2021) | SR50 | K400 | 87.3 | N/A | 51.7 |
| $\rho$MoCo$_{\rho=2}$ (Feich. et al., 2021) | SR50 | K400 | 91.0 | N/A | 54.4 |
| $\rho$BYOL$_{\rho=2}$ (Feich. et al., 2021) | SR50 | K400 | 92.7 | N/A | 55.8 |
| VideoMAE (Tong et al., 2022) | ViT-B | ✗ | 91.3 | 62.6 | 64.3 |
| **EVEREST (Ours)** | ViT-B | ✗ | **93.4** | **68.1** | **64.6** |

Figure 5: **Performance & GFLOPs Comparison** on UCF101 dataset. **(Left)** EVEREST outperforms VideoMAE for both masking ratios (75% and 90%) even at significantly fewer training epochs. **(Right)** Our EVEREST reduces GFLOPs during pre-training and fine-tuning compared to VideoMAE and ST-MAE.

$2i^{th}$ and $(2i + 1)^{th}$ frames are drawn. The model trains video clips with a limited length at each iteration since longer clips require massive computations and memory. Therefore, we remark that information-intensive frame selection is crucial to better capture causality in the arrival video, as the model can observe longer video fragments while avoiding redundant frames.

## 5 EXPERIMENTS

**Experimental settings**   We validate our method on multiple video datasets: *UCF101* (Soomro et al., 2012), *HMDB51* (Kuehne et al., 2011), *Something-Something v2 (SSv2)* (Goyal et al., 2017), *Kinetics-400 (K400)* (Kay et al., 2017) and *Ego4D* (Grauman et al., 2022). We evaluate our information-intensive frame selection strategy during pre-training on *Object State Change Classification (OSCC)*[1] task from Ego4D, containing raw and uncurated people's daily life videos. Given 8-second videos, OSCC classifies whether the object's state has changed by interacting with a camera wearer. Following the same training protocols as VideoMAE (Tong et al., 2022), we pre-train our EVEREST over benchmark datasets *without labels* and report the fine-tuning performance. For a fair comparison, we train both VideoMAE and our EVEREST using the same *one-node* equipped with 8 GPUs. For K400 and SSv2, although VideoMAE trained all models for 1600 and 2400 epochs, respectively, with multi-node GPUs (e.g., 64 V100 GPUs), we train both methods with several scaled ViT backbones for 200 epochs due to quick validation of the scalability. Except for K400, we use ViT-B as a backbone for the other benchmarks. Please see Appendix A for further implementation details.

**EVEREST is memory and computationally efficient while achieving competitive performance** against VideoMAE and K-centered variants. We extensively compare our proposed method against strong VRL baselines. Table 1 shows the results of the supervised and self-supervised methods on K400. Our redundancy-robust token selection, EVEREST, shows comparable performance to VideoMAE (Tong et al., 2022), while achieving *significant* computation and memory reduction during

---

[1]https://github.com/EGO4D/hands-and-objects/tree/main/state-change-localization-classification

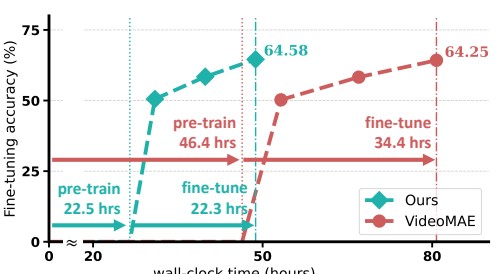

Figure 6: **Accuracy over training time** on SSv2. 4 NVIDIA RTX 3090 GPUs are used. We set $\rho_{pre}$ to 0.3 and 0.8 for pre-train and fine-tuning, respectively.

Table 3: **Memory usage comparison** during the pre-training (one node, A100×8 GPUs) on K400. The memory gaps grow up when increasing batch size and model size. Especially, our EVEREST with ViT-L achieves about 4× **better** efficiency than VideoMAE (634.1GB). *OOM* indicates *out-of-memory*.

| Method | Backbone | Effective Batch Size | | | |
|---|---|---|---|---|---|
| | | 128 | 256 | 512 | 1024 |
| VideoMAE | ViT-S | 71.9 | 117.4 | 233.6 | 443.6 |
| | ViT-B | 258.4 | 486.4 | *OOM* | *OOM* |
| | ViT-L | 353.9 | 634.1 | *OOM* | *OOM* |
| Ours | ViT-S | **31.9** | **59.9** | **95.6** | **173.1** |
| | ViT-B | **60.0** | **91.2** | **151.2** | **273.6** |
| | ViT-L | **117.3** | **164.1** | **258.9** | **428.4** |

Table 4: **Pre-training time & Memory comparison with SoTA MVAs on K400.** We measure the pre-training time for an epoch under a single-node machine equipped with 8×A6000 (48GB) GPUs. We use ViT-B and a batch size of 128. Note that MME (Sun et al., 2023) should pre-compute the optical flow for the entire video data before pre-training. We exclude the time and memory of MME for optical flow computations in the table.

| Method | PT-Time | Memory |
|---|---|---|
| VideoMAE (Tong et al., 2022) | 18m 42s | 150.3 GB |
| MME (Sun et al., 2023) | 10m 15s | 121.2 GB |
| MVD (Wang et al., 2023c) | 51m 55s | 274.9 GB |
| **EVEREST (ours)** | **8m 18s** | **66.3** GB |

Table 5: **EVEREST-Finetuning with other MVAs on K400.** We apply our EVEREST for finetuning the pre-trained models (ViT-B) by SoTA MVAs. We measure the memory usage with the same batch size of 128. While SoTA methods use full tokens during fine-tuning, our EVEREST uses only redundancy-robust tokens that **reduces significant computational cost and memory usage** with comparable accuracy.

| PT-Method | FT-Method | GFLOPs | Memory | Top-1 |
|---|---|---|---|---|
| VideoMAE | Full-token | 180.5 | 362.5 GB | 81.5 |
| VideoMAE | **EVEREST** | 98.1 | 178.4 GB | 81.6 |
| MME | Full-token | 180.5 | 362.5 GB | 81.8 |
| MME | **EVEREST** | 98.1 | 178.4 GB | 82.0 |
| MVD | Full-token | 180.5 | 362.5 GB | 83.4 |
| MVD | **EVEREST** | 98.1 | 178.4 GB | 82.8 |

both pre-training and fine-tuning. Specifically, for ViT-S, EVEREST can reduce computational costs by 45.7% and 48.9%, respectively. It's also worth noting that EVEREST using ViT-L is more than $4\times$ more memory efficient than VideoMAE (e.g., 164.1 GB vs. 634.1 GB). Meanwhile, Table 2 shows that EVEREST achieves superior fine-tuning performance against recent VRL methods across UCF101 and HMDB. Specifically, EVEREST outperforms the best baseline, VideoMAE, by $2.1\%p \uparrow$ on UCF101 and $3.2\%p \uparrow$ on HMDB51. Also, we visualize the convergence plot of EVEREST on UCF101 in Figues 5 Left. By pre-training at only 800 epochs, our EVEREST surpasses the fine-tuning accuracy of VideoMAE trained at 3200 epochs. That is, EVEREST reaches on-par performance with VideoMAE by using only ∼**14% of total computational costs**. We also against variants of a strong motion-based token selection method, K-centered patch sampling, with the modified vision transformer for video learning, including XViT(Bulat et al., 2021), Mformer (Patrick et al., 2021), and TSformer (Bertasius et al., 2021). Our proposed method also surpasses these motion-based video understanding methods by significant margins in terms of Top-1 accuracy and GFLOPs.

**EVEREST is highly beneficial for model deployment.** To validate the efficiency of our EVEREST in terms of memory usage, we measure the memory consumption over multiple batch sizes and architecture scales during the pre-training phase on K400. We use one node equipped with 8×A100 (80GB) GPUs and compare EVEREST with VideoMAE. As shown in Table 3, VideoMAE shows *out-of-memory* when using a batch size of 512 and 1,024. Regarding that VideoMAE used the batch size of 1,024 in the original paper (Tong et al., 2022), VideoMAE inevitably has to lower the batch size to run training in a one-node environment, resulting in increased training time. However, the proposed redundancy-robust token selection approach allows to drastically reduce the memory usage, which means that it can be trained using only a one-node environment. We further compare pre-training and fine-tuning budgets with those of the state-of-the-art MVAs such as MME (Sun et al., 2023) and MVD (Wang et al., 2023c). In Table 4, when pre-training, EVEREST requires only 44%, 24%, and 55% of memory than VideoMAE, MME and MVD, respectively. Due to the generality of our EVEREST, we can apply EVEREST to the finetuning phase by prunning redundant tokens. As shown in Table 5, compared to other methods using full tokens, our EVEREST reducing computation and memory requirements while acheiving comparable accuracy.

**EVEREST is also significantly rapid for pre-training compared to SoTA VMA baselines.** To evaluate the wall-clock time efficiency, we compare pre-training time (PT-Time) with state-of-the-art

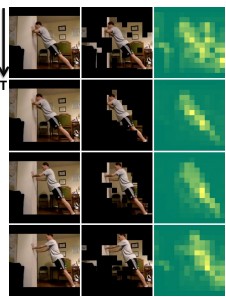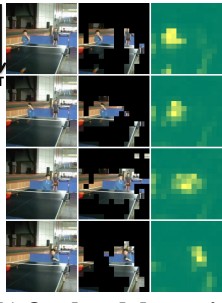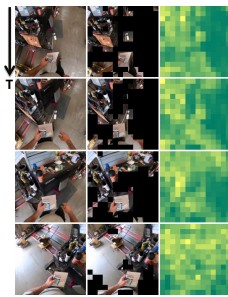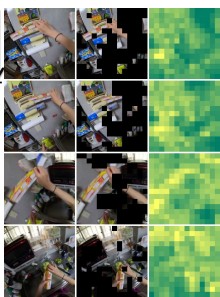

**(a) fixed and static**          **(b) fixed and dynamic**          **(c) changing and messy**

Figure 7: **Examples of Redundancy-robust Token selection.** We visualize the original video frames (left), Redundancy-Robust masking (middle), and obtained importance heatmaps (right) on UCF101 (**(a)** and **(b)**, $\rho_{pre}$=0.3) and Ego4D (**(c)**, $\rho_{pre}$=0.5). Our ReRo token selection successfully captures both narrow/concentrated motion information in curated videos **(a,b)** and distributed/multi-object motion cues in egocentric videos **(c)**.

MVAs by using the same batch size on K400. As shown in Table 4, EVEREST requires only 45%, 16%, and 81% of pre-training time than VideoMAE, MME, and MVD, respectively. Note that we didn't reflect the heavy computation burdens of MME for extracting HOG and optical flow of all input video data before training. Similarly, the whole training time, including both pre-training and fine-tuning, can be drastically reduced, as shown in Figues 6. We note that along with memory efficiency in our EVEREST, we can maximally enjoy remarkably accelerated pre-training and fine-tuning phases given resource-constrained environments than the cases of MVA methods.

**EVEREST has a potential for estimating motion information.** We provide masked input examples with their importance heatmaps through our proposed redundancy-robust masking in Figues 7. Our masking strategy enables the model to capture the most informative tokens containing static and dynamic moving of objects (e.g., *doing wall push up* (Figues 7 (a)) and *playing ping pong* (Figues 7 (b))). Furthermore, our masking strategy captures informative objects even with the rapid change of the view in the first-person videos, as it masks out objects which are crucial for understanding the camera wearers' attention (Figues 7 (c)), while not attending to backgrounds such as walls and floors.

**EVEREST adaptively finds information-dense frames from egocentric videos.** Real-world videos include many redundant scenes over a long time, the VRL model may waste time and computations by learning on meaningless frames, which also can lead to poor local optima due to bias and catastrophic forgetting. To overcome the limitation, we adopt the rate of redundancy-robust sampling $\alpha$ to focus on the frames with larger motions across frames as illus-

| Method | Modality | Accuracy (%) |
|---|---|---|
| Egocentric VLP (Lin et al., 2022) | V+T | 73.9 |
| SViT (Escobar et al., 2022) | V | 69.8 |
| TarHeels (Islam & Bertasius, 2022) | V | 70.8 |
| **EVEREST (Ours)** | V | **76.2** (+5.4%p) |

Table 6: **Comparison with public SoTA methods for OSCC on Ego4D val set.** We set our frame selection ratio $\alpha = 1.5$. Pre-trained data modalities from the OSCC dataset 'V' and 'T' refers to visual and text, respectively.

trated in Figues 3. By constructing the given videos to have fluent motion information, our model focuses more on learning the core part of the video. We set the default $\alpha$ to 1.5 and report the effect of $\alpha$ in Appendix C. We quantitatively compare our method to recent SoTA methods using OSCC task on Ego4D in Table 6. We pre-train OSCC without labels and fine-tune it to classify whether the object's state has changed. With only visual information, we outperform the previous SoTA method Egocentric VLP (Lin et al., 2022), which uses visual and text information, by 2.3%p ↑.

## 6 CONCLUSION

From the insight that not all video tokens are equally informative, we propose a simple yet efficient parameter-free token and frame selection method for masked video autoencoder. We adaptively select the crucial redundancy-robust tokens based on significant spatiotemporal changes in state and only train them, drastically reducing memory allocation and computational cost. In addition, we propose a frame selection technique to construct input video data by sampling frames with a probability proportional to the degree of their occupancy of adaptively obtained tokens. This is useful for video representation learning with uncurated videos containing a number of redundant frames. The experimental results show that our method is significantly more efficient in computations, memory, and training time than the baselines. We believe our method could help democritizing (Seger et al., 2023) video-related research requiring immense computation budgets.

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

# Appendix

**Organization**    The supplementary file is organized as follows: Firstly, we explain the implementation details according to tasks in Appendix A and perform the distance function ablation study for redundancy-robust masking during pre-training in Appendix B. We provide additional experimental results for our EVEREST in Appendix C. Next, we visualize more examples from EVEREST in Appendix D. Finally, we provide a discussion on broader impacts and limitations of our work in Appendix E.

## A    IMPLEMENTATION DETAILS

Table 7: **Pre-training settings for K400, SSv2, UCF101, HMDB51 and OSCC.**

|  | K400 | SSv2 | UCF101 | HDMB51 | OSCC |
|---|---|---|---|---|---|
| optimizer | | | AdamW | | |
| optimizer momentum | | $\beta_1, \beta_2 = 0.9, 0.95$ Chen et al. (2020) | | | |
| base learning rate | 1.5e-4(S,L), 3e-4(B) | 1.5e-4 | 1e-3 | 1e-3 | 1e-3 |
| weight decay | | | 0.05 | | |
| learning rate schedule | | Cosine decay Loshchilov & Hutter (2016) | | | |
| flip augmentation | yes | no | yes | yes | yes |
| augmentation | | | MultiScaleCrop | | |
| batch size | 1024(S,B), 512(L) | 256 | 192 | 192 | 256 |
| warmup epochs | 40 | 40 | 40 | 40 | 20 |
| $\rho_{pre}$ | 0.3 | 0.3 | 0.3 | 0.3 | 0.5 |
| sampling stride | 4 | 2 | 4 | 2 | 4 |
| total epochs | 200 | 200 | 3200 | 4800 | 400 |

Table 8: **Fine-tuning settings for K400, SSv2, UCF101, HMDB51 and OSCC.**

|  | K400 | SSv2 | UCF101 | HDMB51 | OSCC |
|---|---|---|---|---|---|
| optimizer | | | AdamW | | |
| optimizer momentum | | $\beta_1, \beta_2 = 0.9, 0.999$ | | | |
| weight decay | | | 0.05 | | |
| learning rate schedule | | | Cosine decay | | |
| warmup epochs | | | 5 | | |
| laer-wise lr decay | | 0.75 Bao et al. (2021) | | | |
| flip augmentation | yes | no | yes | yes | yes |
| RandAug | | $(9, 0.5)$ Cubuk et al. (2020) | | | |
| label smoothing | | 0.1 Szegedy et al. (2016) | | | |
| drop path | | | 0.1 | | |
| base learning rate | 5e-4(S), 1e-3(B), 2e-3(L) | 5e-4 | 1e-3 | 1e-3 | 1e-4 |
| batch size | 384(S,B), 128(L) | 48 | 128 | 128 | 32 |
| $\rho_{pre}$ | 0.6 | 0.8 | 0.6 | 0.6 | - |
| sampling stride | 4 | - | 4 | 2 | 10 |
| total epochs | 75(S), 35(B,L) | 50 | 100 | 50 | 30 |
| multi-view | 2×3 | 2×3 | 5×3 | 10×3 | 2×3 |

We validate our EVEREST on five video datasets: *K400* (Kay et al., 2017), *SSv2* (Goyal et al., 2017), *UCF101* (Soomro et al., 2012), *HMDB51* (Kuehne et al., 2011), and *Ego4D* (Grauman et al., 2022). We provide the hyperparameter setup for pre-training and fine-tuning in Table 7 and Table 8, respectively. During fine-tuning, we follow segment-based sampling (Wang et al., 2018) on SSv2. As we mentioned in Section 4.1 of the main paper, we adopt the same masking ratio with VideoMAE (Tong et al., 2022) and ST-MAE (Feichtenhofer et al., 2022) for masking input video, but we only reconstruct $(\rho_{pre} \times 100)\%$ of input video tokens during pre-training. And in fine-tuning, our model only takes $(\rho_{pre} \times 100)\%$ of input video tokens. We follow the linear learning rate scheduling of Tong et al. (2022) and Feichtenhofer et al. (2022), $lr = base\_learning\_rate \times batch\_size/256$. Overall implementation of our method is built upon VideoMAE[2](Tong et al., 2022).

---

[2]https://github.com/MCG-NJU/VideoMAE

**Video Action Recognition** We extensively evaluated our method on multiple benchmark datasets for the video action recognition task, including K400, SSv2, UCF101, and HMDB51. For inference, we adapt common multi-view testing with T clips × three crops. That is, the model takes T temporal clips with three spatial crops to cover the overall length and space of the video. Then, we measure the average performance of all views.

**Object State Change Classification (OSCC)** The OSCC dataset is the subset of the Ego4d dataset, consisting of 41.1k/21.2k train/val 8-second videos. Note that, as the Ego4D dataset shows the characteristic of having motion cues that are distributed and containing multi objects in Figues 7 (c) of the main paper, we adopt $\rho_{pre} = 0.5$. We train 400 and 30 epochs with a fixed sampling ratio of 4 and 10 in the pre-training and fine-tuning phases, respectively. We use $\alpha = 1.5$ only in pre-training.

## B    COMPARING FUNCTIONS FOR COMPUTING TOKEN EMBEDDING DISTANCE

We analyze the effect of adopting different distance functions: L1 and L2 distance, negative cosine similarity, and negative CKA (Cortes et al., 2012). We train ViT-B on HMDB51 for 200 and 50 epochs, respectively, only varying the distance function in pre-training as shown in Table 9. As shown, L1 and L2 outperform the alternatives. Therefore, we default to the L2 distance to measure the disparity between token embeddings in temporally adjacent frames.

Table 9: Fine-tuning results on HMDB51 measured by varying the distance function in pre-training.

| Distance Function | accuracy(%) |
|---|---|
| negative cosine | 33.53 |
| negative CKA | 34.58 |
| L1 | 42.22 |
| L2 | **42.81** |

## C    ADDITIONAL EXPERIMENTAL RESULTS

Table 10: **Fine-tuning performance of EVEREST on K400 using the VideoMAE pre-trained model**. We adopt our redundancy-robust token selection method for fine-tuning K400. We use the pre-trained weights by VideoMAE for 1600 epochs and Vanilla indicates a standard fine-tuning of the video action recognition task leveraging all visual tokens.

| Pre-training | Fine-tuning | Backbone | GFLOPs | Accuracy (%) |
|---|---|---|---|---|
| | Vanilla | ViT-S | 57.0 | **79.0** |
| | **+EVEREST (ours)** | ViT-S | **29.1** (↓ 48.9%) | **78.8** |
| VideoMAE | Vanilla | ViT-B | 180.5 | **81.5** |
| | **+EVEREST (ours)** | ViT-B | **98.1** (↓ 45.7%) | **81.6** |
| | Vanilla | ViT-L | 597.2 | **85.2** |
| | **+EVEREST (ours)** | ViT-L | **330.0** (↓ 44.7%) | **84.8** |

**EVEREST fine-tuning using the pre-trained VideoMAE on K400.** We also show the transferability of our redundancy-robust token selection by fine-tuning a pre-trained model with a different method, VideoMAE, on the K400 dataset. We borrow the pre-trained weights in the official repository[3]. Table 10 shows the results with different ViT backbones. Even using publicly available pre-trained models from VideoMAE, our EVEREST show competitive fine-tuning performance compared to the baseline while significantly reducing the computational cost. These results demonstrate that EVEREST can seamlessly adopt an otherwise pre-trained backbone.

Table 11: Fine-tuning performance on UCF101 measured by a varying number of frames.

| Method | # of frames | | GFLOPs | | Accuracy (%) |
|---|---|---|---|---|---|
| | pre-training | fine-tuning | pre-training | fine-tuning | |
| **VideoMAE** | 16 | 16 | 35.48 | 180.5 | 90.80 |
| **EVEREST (Ours)** | 16 | 16 | **19.81** | **98.1** | **93.39** |
| **EVEREST (Ours)** | **24** | **24** | **30.65** | **159.4** | **94.42** |

---

[3]https://github.com/MCG-NJU/VideoMAE/blob/main/MODEL_ZOO.md

**Increasing frame lengths from reduced memory of our EVEREST**   As shown in Figues 1 in the main paper, our EVEREST drastically saves the computational cost in pre-training and fine-tuning. In Table 11, we analyze to compensate for the reduced memory occupancy by increasing the number of input frames from 16 to 24 in pre-training and fine-tuning to observe the results in case of having similar computational costs with VideoMAE. We achieve 94.42% accuracy on UCF101 while still having relatively lower GFLOPs than VideoMAE.

Table 12: Ablation study of on-the-fly information-intensive frame selection of EVEREST.

| $\alpha$ | Acc. (%) |
|---|---|
| - | 73.17 |
| 1.3 | 73.54 |
| 1.5 | **73.85** |
| 1.8 | 73.79 |
| 2.0 | 73.80 |

Table 13: Transferability comparison between VideoMAE and EVEREST on K400 and SSv2 to UCF101 and HMDB51 with ViT-S and ViT-B.

| Method | backbone | Pre-train Dataset | Top1 Acc.(%) UCF101 | HMDB51 |
|---|---|---|---|---|
| VideoMAE | ViT-S | K400 | 84.2 | 54.2 |
| EVEREST | ViT-S | K400 | **89.2** (↑ 5.0%) | **61.4** (↑ 7.2%) |
| VideoMAE | ViT-B | SSv2 | 88.7 | 60.9 |
| EVEREST | ViT-B | SSv2 | **92.2** (↑ 3.5%) | **64.6** (↑ 3.7%) |

**The ratio of information-intensive frame selection**   We conduct an ablation study for the rate of information-intensive frame sampling $\alpha$. We pre-train our model 100 epochs on the OSCC task by varying $\alpha$ from *not using (-)* to *2.0*. Interestingly, as shown in Table 12, the performance with $\alpha$ outperforms baselines (-). And we default $\alpha = 1.5$ for experiments, which shows the best performance than others.

**Transferability of our EVEREST**   We also measure the fine-tuning performance on UCF101 and HMDB51 using larger pre-training datasets in Table 13. We perform fine-tuning experiments with 200 epochs of pre-trained models from K400 and SSv2 in Table 1 and Table 2. Impressively, our method gains a substantial performance enhancement over all experiments compared with VideoMAE.

Table 14: **The impact of the number of Conv3d layers for capturing Redundancy-Robust Token Selection.**

| # of consecutive Conv3d embedding layers | layer1 kernel size | layer2 kernel size | layer3 kernel size | # of params. (M) | Memory usage (GB) | GFLOPs | Fine-tuning accuracy (%) |
|---|---|---|---|---|---|---|---|
| 1 | $2 \times 16 \times 16$ | N/A | N/A | **94.2** / 86.3 | **9.2** / 17.6 | **19.8 / 98.1** | **68.1** |
| 2 | $2 \times 4 \times 4$ | $1 \times 4 \times 4$ | N/A | 102.5 / 94.6 | 11.9 / **17.5** | 34.6 / 112.9 | 66.7 |
| 3 | $2 \times 4 \times 4$ | $1 \times 2 \times 2$ | $1 \times 2 \times 2$ | 97.8 / 89.9 | 11.3 / **17.5** | 38.3 / 116.6 | 66.8 |

**The design of the embedding layer**   We perform an ablation study of our proposed EVEREST regarding the embedding layer design. In our default setting, we use a single Conv3d layer. And we adopt deeper neural networks for spatiotemporal embedding in this experiment. Note that we adjust the kernel size of each layer to maintain the output dimension. As shown in Table 14, stacking more layers for input embedding achieves lower performance than the default setting (i.e., # of consecutive Conv3D embedding layer = 1) on the HMDB51 dataset. Even they require substantial additional training costs in terms of the number of parameters, GFLOPs, and memory usage in most cases of pre-training and fine-tuning.

**The importance of selecting tokens containing rich motion information**   We select to learn temporally changing tokens based on the distance between token embeddings in adjacent frames, thereby, the tokens farther away from adjacent frames contain less redundant information. As shown in Table 15, when we reversely select near-distance tokens in pre-training and fine-tuning, it severely decreases the accuracy, and the performance is the worst if the model selects tokens via reverse strategies in both phases.

Table 15: **The impact of informative-token selection.** It shows better performances both in pre-training and fine-tuning when the model prioritizes selection with the most far-distance (descending) embedded tokens rather than near-distance (ascending) tokens.

| Method | ReRo Masking Pre-train | Fine-tune | Fine-tuning accuracy (%) |
|---|---|---|---|
| EVEREST | ascending | ascending | 60.93 |
| | ascending | descending | 73.49 |
| | descending | ascending | 75.60 |
| | descending | descending | **91.56** |

Table 16: **The ratio of our ReRo Masking at pre-training (Left) and fine-tuning (Right)** on UCF101. We highlight the default redundancy-robust masking rate ($\rho_{pre}$) as red texts. We basically pre-train our model 800 epochs, but also report the results with 3,200 pre-training epochs following VideoMAE, denoted by [†]. [*] and [**] denotes the results with 75% and 90% masking, respectively.

| Method | $\rho_{pre}$ | top-1 | GFLOPs |
|---|---|---|---|
| MAE | - | - | 46.10 |
| VideoMAE[†] | - | 91.25[*] | 57.50 |
| | - | 90.80[**] | 35.48 |
| | 0.5 | 90.91 | 23.43 |
| Ours | 0.4 | 91.51 | 21.54 |
| | 0.3 | **91.56** | 19.81 |
| | 0.2 | 90.80 | **18.22** |
| Ours[†] | 0.3 | **93.39** | 19.81 |

| Method | $\rho_{pre}$ | top-1 | GFLOPs |
|---|---|---|---|
| MAE | - | - | 180.6 |
| VideoMAE[†] | - | 91.25 | 180.5 |
| | 1.0 | 89.71 | 180.5 |
| | 0.8 | 90.17 | 137.5 |
| Ours | 0.7 | 91.25 | 117.3 |
| | 0.6 | **91.56** | 98.1 |
| | 0.5 | 91.48 | **79.8** |
| Ours[†] | 0.6 | **93.39** | 98.1 |

Table 17: **Masking strategy comparison.** To compare with EVEREST, we conduct experiments adopting a frame-wise masking strategy with 400 and 100 pre-training epochs and 100 and 10 fine-tuning epochs on UCF101 and OSCC, respectively. For a fair comparison, the frame-wise masking strategy adopts the same $\rho_{pre}$ of **EVEREST**.

| PT-Making | FT-Masking | Top-1 | |
|---|---|---|---|
| | | UCF101 | OSCC |
| Frame-wise | - | 68.5 | 69.8 |
| Frame-wise | Frame-wise | 72.4 | 69.2 |
| **EVEREST** | **EVEREST** | 89.7 | 73.9 |

**Ablation study on redundancy-robust token selection** One of our major contributions is the significantly enhanced computational efficiency during VRL, as we process a few latent vectors in the decoder to reconstruct only the motion-activated tokens in the given videos according to the redundancy-robust masking ratio $\rho_{pre}$ (Please see Section 4.1). We set $\rho_{pre}$ to $0.3$ so that our EVEREST reconstructs the $30\%$ of the essential spatiotemporal regions focusing on objects' movements and behaviors, from a sparsified video clip, which reduces $65.5\%$ of GFLOPs at the pre-training phase (Please see Figues 5 Right and Table 16 Left). Additionally, as reported in Table 16 Right, our approach can reduce the computational overhead at the fine-tuning phase, which is practically useful when transferring the learned representation learning model to downstream video tasks. Next, We emphasize that redundancy-robust masking is applicable not only in the pre-training phase, but also in performing video-based downstream tasks. We process only $60\%$ of tokens ($\rho_{pre} = 0.6$) of the given video during fine-tuning. Surprisingly, our redundancy-robust masking for the downstream tasks gains increased performance than our variant without masking on the fine-tuning tasks ($\rho_{pre} = 1.0$) using only about $55\%$ of GFLOPs, as shown in Table 16 Right. The results support our hypothesis that video data often contain redundant information and our selective video learning using the proposed masking strategy successfully captures essential space-time regions in video inputs, allowing us to focus more on learning spatiotemporally meaningful features. The fine-tuning masking ratio is higher than the ratio at the pre-training stage, showing that the model uses more information to fully exploit the task-relevant cues from the given videos than pre-training, which aims to obtain general information.

## D    VISUALIZATION OF REDUNDANCY-ROBUST MASKING AND SAMPLING

As shown in Figues 7 of the main paper, our masking method successfully captures the most informative space of each frame in various conditions. In Figues 8, we visualize masking strategies of other strong baselines, ST-MAE (Feichtenhofer et al., 2022) and VideoMAE (Tong et al., 2022). While these two methods sample masked tokens based on random probability without taking into account semantic motion information, our proposed method successfully captures temporally significant localized regions in the video. We further visualize our masking results in Figues 9 for deeper understanding. The upper sample is similar in Figues 7 (b) but changes the $\rho_{pre}$ from 0.5 to 0.15. The masked result shows that our masking strategy could be better and save more computational

costs if the view is fixed and the moving object is few. The bottom sample show that our masking strategy wrongly captured the blue line of each frame as informative tokens when $\rho_{pre}$ is relatively large(=0.5). However, when $\rho_{pre}$ is relatively small(=0.25, 0.15), it ignores the blue line of each frame and concentrates more on moving objects in the given video. In Figues 10, We show selected frames via our adaptive frame sampling. Our method uses redundancy-robust token selection to draw frames to learn in an online manner based on the proportion that contains important tokens containing fluent motion information or significant change of temporal states. This strategy allows the model to take diverse frames from given videos while discarding redundant or low-information frames.

## E  BROADER IMPACTS AND LIMITATIONS

Our work has a positive broader societal impact leading to environment-friendly AI research by drastically reducing energy and carbon costs in the training and inference phases. Current methods in the video understanding field consume tremendous monetary and environmental costs, and we believe our work contributes to overcoming this limited feasibility of video-based research fields by enabling single-node training with larger batch sizes using a large backbone, whereas existing VRL methods require immense memory occupancy and suffer from deploying models in the same setup due to *out-of-memory*. We also aim to enhance the accessibility of those cutting-edge video models by employing practical resources on a single node and pursuing eco-friendly AI research.

On the other hand, while our EVEREST achieves impressive performance improvements along with drastic memory, computation, and training time reduction over multiple benchmark datasets/tasks with different backbone sizes, the improvement in model accuracy for the action recognition task on the K400 dataset becomes relatively small for huge backbones such as ViT-L. We speculate that this is due to their better ability of them to encode spatiotemporal representations into higher-dimensional latent spaces. In addition, our method focuses on capturing short-term motion changes in videos by measuring the temporal disparity between adjacent frames. Although we validated the effectiveness of our method on egocentric videos focusing on temporal semantics or causal interactions, the effect of our method on tasks of long-term episodic memory or the cameraman's intention/interactions under a long horizon view was not thoroughly analyzed in this study. We leave it to future research to analyze and address these limitations.

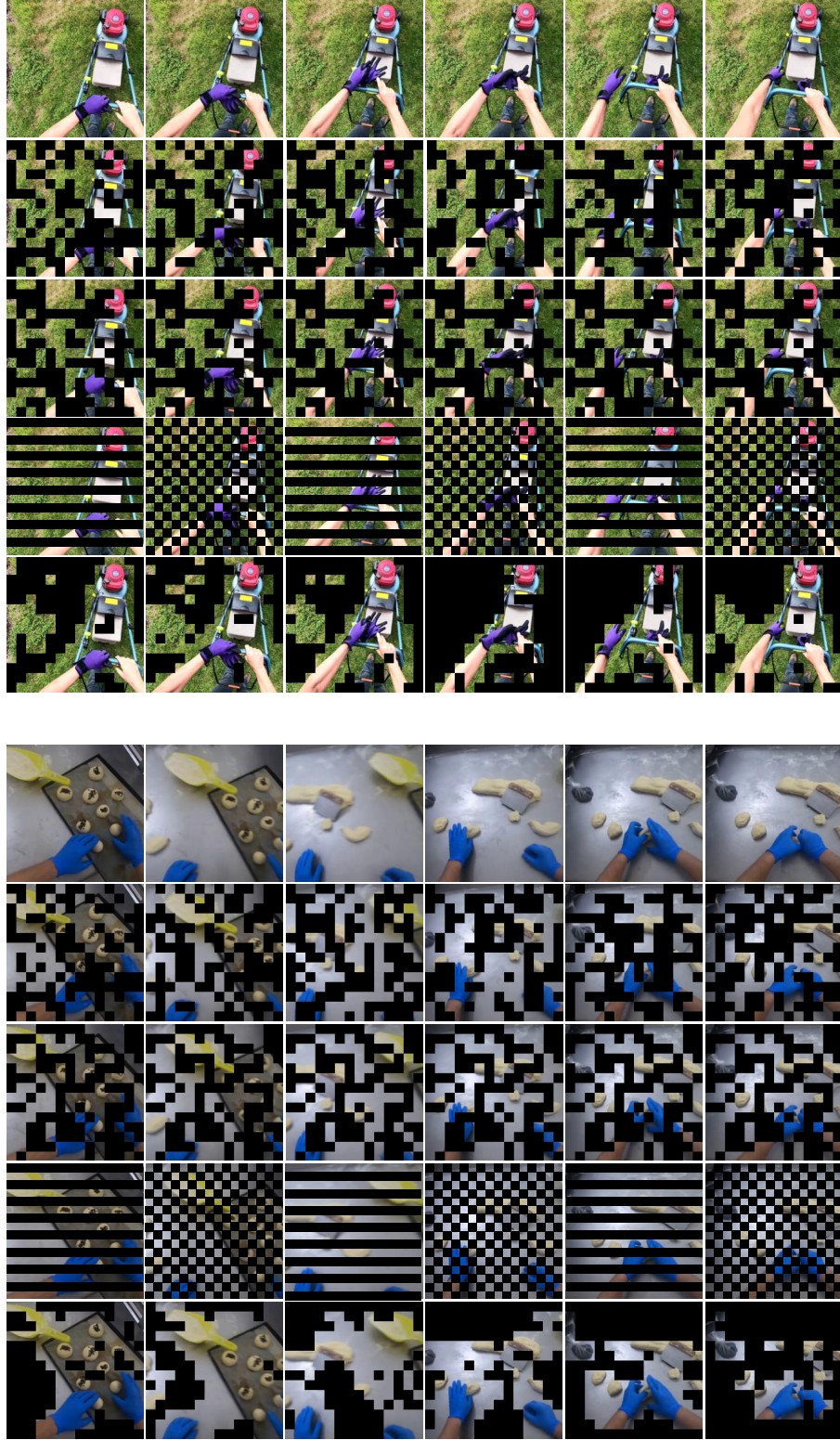

Figure 8: **Examples of masking strategies.** We visualize the masking strategy of ST-MAE (Feichtenhofer et al., 2022) (random), VideoMAE (Tong et al., 2022) (space-only), VideoMAE V2 (Wang et al., 2023a) (running cell (Qing et al., 2023)), and our EVEREST on Ego4D dataset. From the top, each row indicates the original frames, MAE masking, VideoMAE masking, VideoMAE V2 masking, and our proposed method, respectively. We set $\rho_{pre}$ to 0.5.

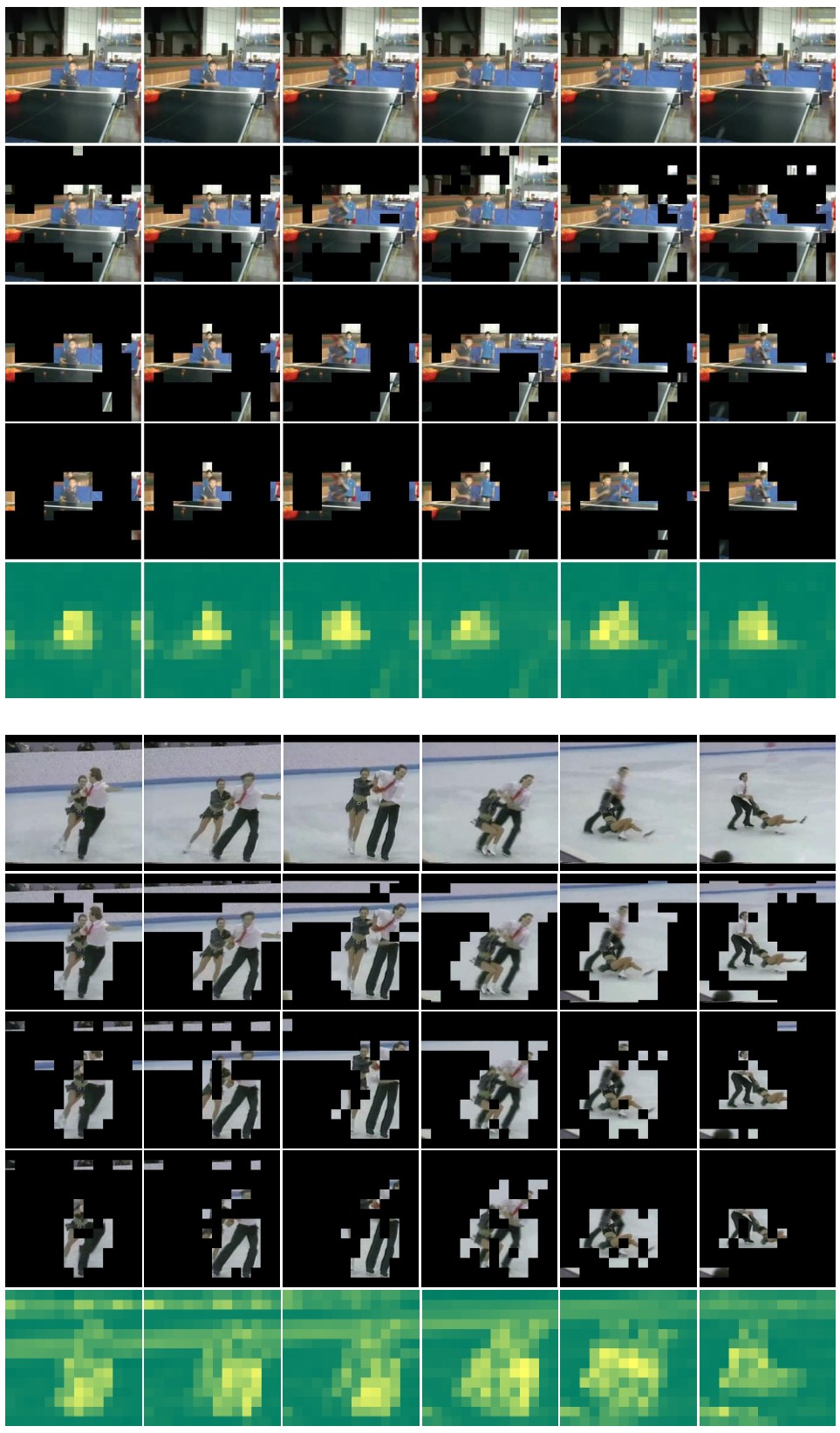

Figure 9: **More examples of Redundancy-robust token selection.** We show the original video frames in the first row, Redundancy-robust masking results in the second, third, and fourth row by varying the $\rho_{pre}$ to 0.5, 0.25, and 0.15, respectively, and obtained importance heatmaps in the last row on UCF101 (Soomro et al., 2012) dataset.

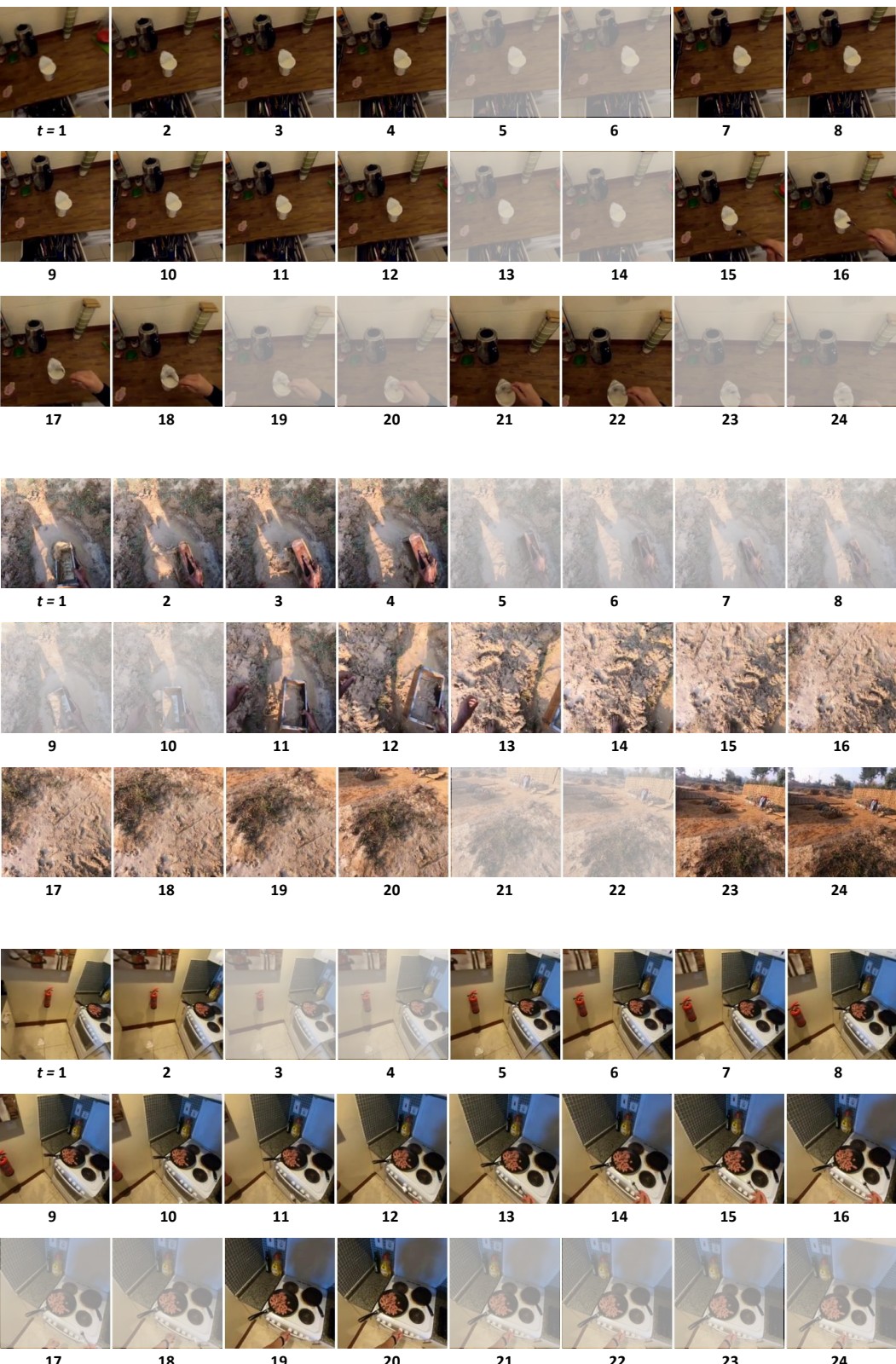

Figure 10: **Examples of on-the-fly information-intensive frame sampling based on our EVEREST.** Among 24 frames in the given video clip, our EVEREST adaptively selects frames containing rich motion information based on temporal correlation. Non-blurred frames are sampled through our method and used in the pre-training phase.

