# OpenReview forum: "EVEREST: Efficient Masked Video Autoencoder by Removing Redundant Spatiotemporal Tokens"
_ICLR.cc/2024/Conference — Submitted to ICLR 2024_

### Official Review · Reviewer_viqa · 2023-10-31

**Soundness:** 4 excellent
**Presentation:** 4 excellent
**Contribution:** 3 good
**Rating:** 5
**Confidence:** 5

**Summary:**

The authors propose a new masking strategy for masked autoencoders for video representation learning. Instead of using random and tube masking which may select tokens from uninformative regions, the authors exploit the unequal information density among the patches by computing the Euclidean distance between the same location across adjacent frames. Moreover, they propose information-intensive frame selection which selects informative frames while discarding the not-so useful ones to learn diverse and robust representations. The new masking strategy is computationally less expensive than the recent masking strategies and it can also be plugged in during the fine-tuning. Experiments on various video datasets show the effectiveness of the masking strategy.

**Strengths:**

- The motivation behind the new masking strategy is well written and clear.
- The paper presents a simple yet effective masking strategy that computes the distance between a patch embedding in one frame with the embedding of the same patch location in the next frame, to determine if there is high information or redundancy there.
- It only reconstructs the high information tokens by sampling a few tokens based on $ρ_{pos}$ ratio.
- Allows the pre-training to be computationally and memory wise less expensive. For example, For a ViT-L backbone with a batch size of 256, method achieves about 4× less memory consumption than VideoMAE. Evaluation was performed using PT and FT GFlops, and memory usage in terms of GB.
- Masking can be used during fine-tuning stage makes it better and faster than VideoMAE.
- Extensive experiments on UCF101, HMDB-51, SSv2, K400, Ego4D.

**Weaknesses:**

- How would the approach work if there is camera motion? I guess it perform adequately for a fixed camera.
- The results on SSv2 and K400 was compared only for 200 epochs. Results on K400 and SSv2 with more epochs?
- Looks like the results of VideoMAE and EVEREST of ViT-L at 200 epochs are about the same.
- On smaller dataset and smaller ViT, the method performs well, but its marginally about the same as VideoMAE on SSv2 with much larger models.
- The paper mostly compare with VideoMAE, but there have been multiple developments in MAE for videos. I would encourage the authors to compare the approach with more baselines e,g, AdaMAE [1], MME [2], MVD [3], ST-MAE [4] for videos (using random masking), Omnimae [5].
- It would be great to add a comparison of the masking strategy against multiple masking strategies.
- Although the approach seems to be computationally less expensive, the results are about the same as VideoMAE, sometimes even degrades the performance of pre-trained masked autoencoder methods for videos (c.f, Table 5).

### References:

[1] AdaMAE: Adaptive Masking for Efficient Spatiotemporal Learning with Masked Autoencoders, CVPR 2023.

[2] Masked Motion Encoding for Self-Supervised Video Representation Learning, CVPR 2023.

[3] Masked Video Distillation: Rethinking Masked Feature Modeling for Self-supervised Video Representation, CVPR 2023.

[4] Masked autoencoders as spatiotemporal learners. NeurIPS, 2022.

[5] Omnimae: Single model masked pretraining on images and videos, CVPR 2023.

**Questions:**

Please see my questions in the weakness section.

---

> ### Author Response · Authors · 2023-11-19
>
> Thank you very much for your review and comments. We provide our response in the following. We strongly believe we have clearly addressed all your concerns. Please don’t hesitate to let us know if your concerns/questions are still not clearly resolved. We are open to active discussion and will do our best to address your concerns during the rebuttal period.
>
> >  _**How would the approach work if there is camera motion? I guess it performs adequately for a fixed camera.**_
>
> #### $\rightarrow$ Our method is also effective for egocentric videos, e.g., **Ego4D**, which capture **_dynamic views from camera motion_**. As visualized in Figure 7 (c), our masking strategy captures informative objects even with the rapid change of the view in the first-person videos, as it masks out objects crucial for understanding the camera wearers’ attention while not attending to the background, such as walls and floors. Additionally, we quantitatively compare our method to recent SoTA methods using the OSCC task on Ego4D in Table 6. Further, we emphasize that egocentric videos contain much temporally redundant visual information. Figure 8 reveals that a worker focuses on a grass mower to operate it well, and a person makes cookie dough, where visual scenes include much meaningless visual information, like empty space on the table. Our method automatically discards these noisy and redundant tokens in an online manner without requiring additional dense information, such as optical flows of incoming videos.
>
>
> > **_The results on SSv2 and K400 was compared only for 200 epochs. Results on K400 and SSv2 with more epochs?_**
> ####  $\rightarrow$ We first respectfully hope that you could understand the limited budget of our research group. We want to emphasize this work is on academic paper to cut down the immense scale of video unsupervised learning, rather than industrial paper using extreme resources.
>
> Additionally, We validated the strengths of our proposed EVEREST through extensive experiments on multiple architectures (ViT-S/B/L), benchmarks (UCF101, HMDB51, SSv2, K400, Ego4D), and tasks (action recognition, OSCC) with sufficient analyses in our main paper that consistently show better efficiency and comparable performance to VideoMAE, so we expect our method to also be effective in longer training period settings.
>
>
> And, we politely want to say that **we provided very long training epochs results in our main paper**: We validated our method by training 4800 and 3200 epochs** on the UCF101 and HMDB51 datasets (Please see Table 7), respectively, and achieved superior fine-tuning accuracy along with memory/training time/computational efficiency by a large margin (Please see Table 2 and Figure 5 in the paper).
>
> >  **_Comparable or marginal performance gain compared to VideoMAE_**
> #### $\rightarrow$ We would like to emphasize that the **goal** of this work is to make the masked video modeling (MVM) more **_efficient_** without performance drop rather than just achieving state-of-the-art performance.  Several works [1,2,3] reducing redundant tokens of ViT in the image domain have achieved its efficiency but sacrificed performance drop. However, we elaborate on how to effectively reduce redundant tokens in the spatiotemporal domain and find a way to achieve efficiency without performance drop. Therefore, we would like to highlight that it is meaningful to increase efficiency without losing performance. Furthermore, the computation and memory efficiency of our EVERST enables us to train MVM models using only **single node** with 8 GPUs, which helps the vision community to **lower the barrier to further SSL for video research.**
>
> > **_It would be great to add a comparison of the masking strategy against multiple masking strategies._**
> #### $\rightarrow$  We have already compared multiple masking strategies by visualizing the masked results (Please see Figures 7 and 10.), and it shows our ReRo token selection successfully captures the informative tokens and removes redundant tokens from the given videos.
>
> ---
>
> #### [1] Rao et al. Dynamicvit: Efficient vision transformers with dynamic token sparsification. NeurIPS 2021.
> #### [2] Liang et al. Expediting vision transformers via token reorganizations. ICLR 2022.
> #### [3] Bolya et al. Token merging: Your ViT but faster. ICLR 2023.

---

> ### Author Response · Authors · 2023-11-19
>
> >  **_Compare the approach with more baselines e,g, AdaMAE[1], MME[2], MVD[3], ST-MAE[4] for videos (using random masking), Omnimae[5]_**
>
> #### $\rightarrow$ First, we want to politely emphasize **we already provided the comparison with MME[2] and MVD[3] in Table 4 and 5 in our submission!**
>
> ST-MAE [4] has a very similar architecture to VideoMAE and also performs similarly (For example, VideoMAE outperforms ST-MAE in [3], and ST-MAE shows on-par performance In [2].). So, we use VideoMAE as our counterpart baseline, instead of ST-MAE, given the limited resources.
>
> #### OmniMAE [5] can learn versatile representation compatible with image and video domains. However, they require massive computational costs (similar GFLOPs and Params with MVD [3]) while achieving a **_lower_** action recognition performance than VideoMAE and MVD [3].
>
> #### MVD [3] distills knowledge from pre-trained image and video teachers during pre-training. MVD improves performance on top of VideoMAE due to knowledge transfer from **rich and heavy teacher models**, but **it suffers from nontrivial computational costs and slow training time**. MME [2] predicts the motion trajectory of the masked video contents, which is encoded by HOG and optical flow for modeling the shape and the position of moving objects. MME strategy can capture motion more accurately. Yet, it **requires computing HOG and optical flow** of all input video data before training, resulting in another **heavy computation burden**.
>
> ####  AdaMAE [1] introduces an auxiliary token sampling network to MAE, which estimates categorical distribution over spacetime-patch tokens and then samples higher spatiotemporal tokens from the distribution. Due to the adaptive sampling network, it can reduce the masking ratio from 90% (VideoMAE) to 95% while achieving better performance than VideoMAE. However, its computation reduction rate is not significant and it needs to further train the auxiliary network. However, Unlike MME and AdaMAE, our EVEREST can capture motion information **_in an online manner without an external module or additional training_**. Furthermore, our method can be applied even when **the fine-tuning phase** (all other methods cannot), which demonstrates that our EVEREST is a more general and flexible method. To support this claim, we already showed the finetuning computation and memory usage comparison in **Table 5**.
>
> ---
>
> #### [1] AdaMAE: Adaptive Masking for Efficient Spatiotemporal Learning with Masked Autoencoders, CVPR 2023.
> #### [2] Masked Motion Encoding for Self-Supervised Video Representation Learning, CVPR 2023.
> #### [3] Masked Video Distillation: Rethinking Masked Feature Modeling for Self-supervised Video Representation, CVPR 2023.
> #### [4] Masked autoencoders as spatiotemporal learners. NeurIPS, 2022.
> #### [5] Omnimae: Single model masked pretraining on images and videos, CVPR 2023.

---

> ### Author Response · Authors · 2023-11-21
> **Dear Reviewer viqa - A Gentle Reminder**
>
> Dear Reviewer viqa,
>
> We are sincerely grateful to you for reading our response.
>
> During the rebuttal period,
>
> - We have clarified that our proposed approach can **handle tasks/videos containing dynamic camera motion** very well.
> - We have emphasized that we already provided **very long training epochs** (HMDB51, UCF101), and we really hope you could understand the limited training budgets in our research group. Instead, we verified the efficacy of our proposed method **through extensive empirical evaluations** in terms of architectures, tasks, datasets, and visualizations.
> - We have emphasized that the **primary goal** of this work is to make masked video modeling (MVM) **more efficient without performance drop** rather than just achieving state-of-the-art performance.
> - We have clarified that we **already provided comparisons with multiple masking strategies** by visualizing the masked results.
> - We have emphasized that we **already provided the comparison with MME[2] and MVD[3] in Tables 4 and 5** in our submission as well.
>
> We remain committed to further improving the quality of our paper by addressing any remaining concerns and suggestions where necessary. With that in mind, if you might have any further feedback, please let us know. We would be grateful for the opportunity to address them and make our work a more solid and valuable contribution to the field of video representation learning.
>
> **Also, we would like to kindly suggest your reconsideration of the rating**, if you feel that our work does not have major concerns with respect to evaluation, resources, reproducibility, and ethical considerations. We understand that the criteria for rating a paper can sometimes be subjective; however, we believe that our work can meet the requirements of the above score, given that most of your concerns are effectively addressed and as long as there are no major issues.
>
> We thank you so much for your time and effort in reviewing our paper, and your constructive feedback that has greatly contributed to improving our paper.
>
> Warm Regards,
> Authors

---

> ### Author Response · Authors · 2023-11-22
> **Thank you for your review; Today is the end of the discussion phase.**
>
> Dear Reviewer viqa,
>
> We sincerely appreciate your efforts in reviewing our paper, and your constructive comments. We have clarified all your comments.
>
> As you know, now we have only one day to have interactive discussions. Could you please go over our responses and the revision since the end of the final discussion phase is approaching? Please let us know if there is anything else we need to clarify or provide.
>
> Best,
> Authors

---

> ### Author Response · Authors · 2023-11-23
> **Discussion phase ends within 8 hours**
>
> Dear Reviewer viqa,
>
> We really appreciate your effort in reviewing our submission again. Since the discussion period for ICLR 2024 ends within 8 hours, we politely ask you to read our new responses by any chance. Please understand that we have made our best effort to address your concerns during this period.
>
> Also, we would like to kindly suggest **a reconsideration of the initial rating (reject: 5)**, if you agree that **most concerns raised by the initial review are resolved**. We strongly believe that most of your concerns are effectively addressed as long as there are **no significant issues to be negative**.
>
> We thank you so much for your time and effort in reviewing our paper and for the constructive feedback that has greatly improved it.
>
> Best,
> Authors

---

### Official Review · Reviewer_JSFH · 2023-11-01

**Soundness:** 3 good
**Presentation:** 3 good
**Contribution:** 3 good
**Rating:** 8
**Confidence:** 4

**Summary:**

This paper presents EVEREST, a masked video autoencoder, which proposes token selection and frame selection strategies aimed to focus on motion information and discard redundant spatiotemporal information from the input. This method demonstrates comparable or superior performance to existing baselines such as VideoMAE on several benchmarks while maintaining lower computational and memory costs, both during pre-training and fine-tuning.

**Strengths:**

- EVEREST uses a measure of relative importance to select the most informative token embeddings from each video and learn to only use them during pre-training and fine-tuning, discarding the redundant ones. This can substantially reduce computational costs.

- Instead of employing uniform frame sampling, EVEREST proposes a strategy to sample frames with distinct spatiotemporal features based on the amount of informative token embeddings each of them contains.

- Both of the above strategies lead to relatively more computationally and memory-efficient pre-training and fine-tuning while producing comparable or better downstream performance at different scales of the ViT backbone.

- Ablation studies in the Appendix strengthen the argument in favor of the proposed strategies.

- To the best of my knowledge, the proposed method is novel and would be of interest to the ICLR community.

- The paper is written and organized clearly.

**Weaknesses:**

The paper could benefit from confidence intervals for the reported accuracies in Tables 1 and 2.

**Questions:**

- To confirm my understanding of the sensitivity of EVEREST to the choice of $\rho_{pre}$ in Table 16 in the appendix,  is $\rho_{pre}\cdot \rho_{post}$ always set to 0.1 for a given value of $\rho_{pre}$?

- Is it possible to provide confidence intervals for the results in Tables 1 and 2?

- In equation (3), following the notation in section 4.1., should not $\tilde{k}^n$ be $(\tilde{k}')^n$ since $m^n$ is a mask over $\tau \times [J \cdot \rho_{pre}]$? Also, $\tilde{v}$ is not defined (I assume it’s parts of the video that contain $(\tilde{k}')^n$).

Are authors planning to release the implementation of their method?

Suggestions:
- It would be interesting to see the full curves for VideoMAE (75% and 90%) in Figure 5, not just at 3200 pre-training epochs.

- A comparison to VideoMAE2 could also be interesting.

---

> ### Author Response · Authors · 2023-11-19
>
> Thank you very much for your review and comments. We provide our response in the following. We strongly believe we have clearly addressed all your concerns. Please don’t hesitate to let us know if your concerns/questions are still not clearly resolved. We are open to active discussion and will do our best to address your concerns during the rebuttal period.
>
> ---
>
> > ***W1 & Q2. The paper could benefit from confidence intervals for the reported accuracies in Tables 1 and 2. Is it possible to provide confidence intervals for the results in Tables 1 and 2?***
>
> $\rightarrow$ Thank you for your constructive comments! We respectfully hope that you understand the limited short rebuttal period. We could train our model repeatedly with different random seeds to provide confidence intervals. However, we only have a little time to offer it. Instead, we have provided the implementation code already in the supplementary material, and anyone can easily reproduce it and see similar results in Tables 1 and 2. We will make sure to report the confidence interval for our main table in our final revision.
>
> ---
>
> > ***Q1. Is $\rho_{pre}\cdot\rho_{post}$ always set to 0.1 for a given value of $\rho_{pre}$?***
>
>
> $\rightarrow$ Yes, we conducted an ablation study in Table 16 in the appendix, only varying $\rho_{pre}$ while fixing $\rho_{pre}\cdot\rho_{post}$ to be 0.1.
>
> ---
>
> > ***Q3. In equation (3), following the notation in section 4.1., should not $\widetilde{k}^n$ be $(\widetilde{k}')^n$ since $m^n$ is a mask over $\tau \times [J\cdot\rho_{pre}]$? Also, $\widetilde{v}$ is not defined.***
>
> $\rightarrow$ Sorry for the confusion. We understood the point you were confused about and clarified the explanations of notations in our revision. Here, we briefly correct some notations.
>
> Let $\widetilde{\boldsymbol{\mathit{k}}}$ be the token embeddings of the top-$[\tau\cdot\rho_{pre}]$ highest importance based on Eq. (2) and $\widetilde{\boldsymbol{\mathit{k}}}’$ be the randomly sampled token embeddings from $\widetilde{\boldsymbol{\mathit{k}}}$ with the ratio of $\rho_{post}$. Then, an input forwarded by $f_{\mathcal{W}}$ is $\widetilde{\boldsymbol{\mathit{k}}}’$, and the reconstructed target is $\{\boldsymbol{\mathit{m}}} \otimes {\boldsymbol{\mathit{v}}}$. Please see the revision for details.
>
> ---
>
> > ***Q4. Are authors planning to release the implementation of their method?.***
>
>
> $\rightarrow$ We have already released the code. Please check the supplementary material.
>
> ---
>
> > ***Suggestion 1. Draw the full curves for VideoMAE(75% and 90%) in Figure 5.***
>
> [1] Tong, Song, et al. "Videomae: Masked autoencoders are data-efficient learners for self-supervised video pre-training." NeurIPS 2022
>
> $\rightarrow$ Thanks for suggesting such a great idea. It would show the effectiveness and efficiency of our model clearly. However, [1] only reports the results of 3,200 pre-training epochs in their paper, and we don't have enough time and resources to implement their method during this short rebuttal period. After this rebuttal period, we'll update the full curves in our final revision.

---

> ### Author Response · Authors · 2023-11-19
>
> > ***Suggestion 2. Compare to Videomae v2.***
>
> [2] Wang, Limin, et al. "Videomae v2: Scaling video masked autoencoders with dual masking." CVPR 2023
>
> $\rightarrow$ Thank you for the great suggestion. VideoMAE-V2 simply attaches running cell masking [3] for decoder masking in VideoMAE. Running cell masking is an input-independent ad hoc approach; the model constructs square cells based on the spatial location of the frame (i.e., meta-patch), and then randomly samples tokens from **unselected regions in the previous frame** for masking. That is, **VideoMAE-V2** simply aims to mask different regions for each frame (i.e., mask redundancy) and **cannot capture visual redundancy** according to the input videos. To see clear differences and limitations in running-cell masking used in VideoMAE-V2, **we additionally visualized real masking examples** in **Figure 8** in our latest revision. As shown, while EVEREST can focus on informative spatial-temporal tokens in incoming videos, running-cell masking just avoids the overlap of masks between adjacent masking and **cannot capture visual redundancy ($\neq$ masking redundancy) or semantic importance**.
>
> For quantitative analysis, during the rebuttal period, we have compared the effect of our redundancy-robust token masking with running-cell masking used in MAR [3] and VideoMAE-V2. We pre-trained with these masking strategies and performed conventional full-finetuning. As shown in the table below, **our EVEREST clearly outperforms running-cell masking**, demonstrating our effectiveness against other selection strategies given the same architecture.
>
> | method |UCF101|HMDB51|
> |---------------|------------|-----------|
> | Running cell | 91.0 | 61.4 |
> | EVEREST (Ours)| **93.4** | **68.1**|
>
> [3] Qing et al., MAR: Masked Autoencoders for Efficient Action Recognition, IEEE Transactions on Multimedia, 2023

---

> > ### Comment · Reviewer_JSFH · 2023-11-22
> > **Thank you for your responses**
> >
> > Thank you for the clarifications and additional experiments with adopting running-cell masking in the decoder. I don't have any follow up questions at the moment.

---

> > > ### Author Response · Authors · 2023-11-22
> > >
> > > We are sincerely grateful to you for reading our response, and we are glad that you find our additional experiments with running-cell masking, and now everything is clear.
> > >
> > > Best,
> > > Authors

---

### Official Review · Reviewer_4Vgt · 2023-11-01

**Soundness:** 2 fair
**Presentation:** 2 fair
**Contribution:** 2 fair
**Rating:** 3
**Confidence:** 4

**Summary:**

This paper mainly focuses on improving VideoMAE's training efficiency in both pre-training and fine-tuning stages. Specically, they propose a new token selection method based on similarity between tokens to reduce token reducney. Futhermore,  a fram-seletion method is proposed to enhance representation qualizaty. Experiments are conducted on several video benchmarks. The main baseline used for comparison in this paper are VideoMAE. The computation can be saved by up tp 50% and memory can be saved futher.

**Strengths:**

This paper practically demonstrates redandancy can be further reduced in VideoMAE in which a high mask is applied. The proposed method is simple and straightforward, which should not be considered a big innovation (see weakness) but the simplicity should be appreciated.  Such practical observation is somewhat useful the significance is determined by the effectiveness of the proposed method. I can't say the significance is huge through the current results (see weakness). The evaluation is extensive and I  think the visualization of this paper is pretty good.  Regarding clarity, I would say there is still space to improve since results can be organized better and this paper lacks in-depth analysis of the proposed method.

**Weaknesses:**

1. The novelty of the proposed method is limited. First, I would say the similarity-based token selection method has been studied before.  An important baseline is K-centered Patch Sampling in [1]. The core idea is close, and the method both highly rely on the rank of the similarity matrix.  Second, the conclusion or observation of this paper is somewhat novel but has also been showed in previous work. In particular, the VideoMAE-V2 [2] has already been shown in VideoMAE pretraining and fine-tuning. There is plenty of redundancy, so they introduce dual-masking. The conclusion both show us that the efficiency of training video models can be further improved. Since the high-mask ratio reconstruction (90%~95% in the video) has told us the key of VideoMAE pre-training, which has been discussed very well, the further improvement in efficiency can be viewed as incremental, and such improvement has been shown in [2]. So, I would say the observation is not that novel as well.

2. The significance is limited. Since this is an experimental-based paper. The significance mainly relied on the results or improvement over current state-of-the-art or strong baselines.  First, the performance is compared under a very short pre-training period in Table 1. I am concerned about this since, in the original paper, at least an 800-epoch training should be conducted for a fair comparison. Second, When measuring the wall-clock time memory consumption to show efficiency, different devices are used.  To make an accurate claim, I believe they should be benchmarked on the same device.  Finally,  the proposed method only achieved comparable performance under limited budgets, so "state-of-the-art" performance should be carefully used in the abstract.


3. The presentation can be improved. The ablation study is disorganized since the author uses different datasets and different models, even different devices. I would see a clear ablation for the proposed method is needed for better presentation.  In Table 1, I can not understand the comparison of the computations saving since it only happened when compared with VideoMAE.   I cannot understand Table 3-5 as well since there is insufficient consistency in the author's experiments.

[1] Park, Seong Hyeon, et al. "K-centered patch sampling for efficient video recognition." European Conference on Computer Vision. Cham: Springer Nature Switzerland, 2022.
[2] Wang, Limin, et al. "Videomae v2: Scaling video masked autoencoders with dual masking." Proceedings of the IEEE/CVF Conference on Computer Vision and Pattern Recognition. 2023.

**Questions:**

In addition, I have some questions after reading this paper:
1. How do you choose the hyper-parameters for the proposed token selection method?
2. What is the effectiveness of the proposed frame-sampling method?

---

> ### Author Response · Authors · 2023-11-19
>
> Thank you very much for your review and comments. We provide our response in the following. We strongly believe we have clearly addressed all your concerns. Please don’t hesitate to let us know if your concerns/questions are still not clearly resolved. We are open to active discussion and will do our best to address your concerns during the rebuttal period.
>
> ---
>
> > ***W1-1. The novelty of the proposed method is limited: the similarity-based token selection method has been studied before. An important baseline is K-centered Patch Sampling in [1]. The core idea is close, and the method both highly rely on the rank of the similarity matrix.***
>
> [1] Park, Seong Hyeon, et al. "K-centered patch sampling for efficient video recognition." ECCV 2022
>
> $\rightarrow$ We respectfully disagree that the idea of our method is close to [1], and we believe the reviewer misunderstands our method in some sense - ***we do not compute any form of the similarity matrix***.
>
> K-centered patch sampling [1] maps all patch embeddings in the vector space and sample K patches by **running the greedy farthest point sampling algorithm** (Please see Figure 2 in the [1]). This sampling strategy has a critical limitation in that they cannot avoid sampling redundant patches. Suppose two video patches $A_1$ and $A_2$ are identical, and there is another patch $B$ that is the farthest with $A_1$ and $A_2$. If the sampling algorithm picks $A_1$ (or $A_2$), it will pick $B$ (the most distant from $A$). Next, the algorithm will pick $A_2$ (or $A_1$), which is the farthest from $B$. But, as shown, *selecting the farthest tokens* based on the algorithm [1] doesn't mean *minimizing redundancy*.
>
> Unlike [1], our EVEREST guarantees to prevent redundant sampling with our intuitive yet powerful redundant-robust token selection strategy. Furthermore, we suggest on-the-fly information-intensive frame selection that adaptively determines semantic dense (i.e., informative) frames based on the occupancy of our selected redundancy-robust patches.
>
> Therefore, K-centered [1] and ours seem to share similar intuitions, but our proposed method is much more efficient and practical. This is obviously observed in Table 1 in the submission.
>
> (This is a partial table from Table 1 in the submission.)
> | method | PT-GFLOPs | FT-GFLOPs | Top-1 Acc |
> |---------------|------------|-----------|-----------|
> | K-centered [1] |  67.4| 369.5| 74.9|
> | EVEREST (Ours)| **6.3 ($10.7 \times\downarrow$)**| **29.1 ($12.7 \times\downarrow$)** | **75.9 ($1.0$ %p $\uparrow$)**|
> | EVEREST (Ours)| **21.5 ($3.13 \times\downarrow$)**| **98.1 ($3.77 \times\downarrow$)** | **79.2 ($4.3$ %p $\uparrow$)**|
>
> Our proposed EVEREST surpasses these baselines by significant margins in terms of Top-1 accuracy and computation efficiency (GFLOPs) on the K400 dataset, demonstrating that our approach contains sufficient originality in design, and clearly differs from [1].

---

> > ### Comment · Reviewer_4Vgt · 2023-11-21
> > **An ablation can be more helpful**
> >
> > Thank you for your clarification. Regarding your statement, "As shown, selecting the farthest tokens based on the algorithm [1] doesn't mean minimizing redundancy," it could be strengthened with additional support from an ablation study. Since the comparison in your table lacks fairness, the results presented are without MAE pre-training, and there's a difference in the backbone used. With these disparities, it's challenging to conclusively assert the superior performance of your method over the K-centered approach.

---

> ### Author Response · Authors · 2023-11-19
>
> > ***W1-2. The conclusion or observation of this paper is somewhat novel, but VideoMAE-V2 [2] has reduced redundancy in VideoMAE pretraining and fine-tuning through dual-masking. Since the high-mask ratio reconstruction (90%~95% in the video) has told us the key of VideoMAE pre-training, which has been discussed very well, the further improvement in efficiency can be viewed as incremental.***
>
> [2] Wang, Limin, et al. "Videomae v2: Scaling video masked autoencoders with dual masking." CVPR  2023
>
>
> $\rightarrow$ We want to respectfully correct the reviewers' misbelief on the VideoMAE-V2. First, VideoMAE-V2 simply attaches running cell masking [3] for decoder masking in VideoMAE, which can be understood *incremental*. Running cell masking is an input-independent ad hoc approach; the model constructs square cells based on the spatial location of the frame (i.e., meta-patch), and then randomly samples tokens from **unselected regions in the previous frame** for masking. That is, **VideoMAE-V2** simply aims to mask different regions for each frame (i.e., mask redundancy) and **cannot capture visual redundancy** according to the input videos. To see clear differences and limitations in running-cell masking used in VideoMAE-V2, **we additionally visualized real masking examples** in **Figure 8** in our latest revision. As shown, while EVEREST can focus on informative spatial-temporal tokens in incoming videos, running-cell masking just avoids the overlap of masks between adjacent masking and **cannot capture visual redundancy ($\neq$ masking redundancy) or semantic importance**.
>
> For quantitative analysis, during the rebuttal period, we have compared the effect of our redundancy-robust token masking with running-cell masking used in MAR [3] and VideoMAE-V2. We pre-trained with these masking strategies and performed conventional full-finetuning. As shown in the table below, **our EVEREST clearly outperforms running-cell masking**, demonstrating our effectiveness against other selection strategies given the same architecture.
>
> | method |UCF101|HMDB51|
> |---------------|------------|-----------|
> | Running cell | 91.0 | 61.4 |
> | EVEREST (Ours)| **93.4** | **68.1**|
>
>
> In the end, we politely disagree with the Reviewer's claim that further improvement in efficiency beyond a high masking ratio in VideoMAE can be viewed as incremental. **Efficient masking and reducing the computational cost are undoubtedly crucial, practical, and promising research topics and remain an open problem and nontrivial to solve.**
>
> [3] Qing et al., MAR: Masked Autoencoders for Efficient Action Recognition, IEEE Transactions on Multimedia, 2023
>
>
> ---
>
> > ***W2. The significance is limited since this is an experimental-based paper.***
>
> $\rightarrow$ We politely disagree with your claim that the significance is limited due to our main contributions being from the experiments, yet strongly believe that **a simple yet effective method could make a meaningful contribution to the research community.** Many of the high-impact work such as dropout [1], prompt-tuning [2], chain-of-thought prompting [3], and parameter-efficient Transfer Learning Techniques (Adapter [4], LoRA [5], etc.) are all impressively simple, but the simplicity is rather an advantage as it promotes more widespread adaptation of the method. On the other hand, there are many complicated and esoteric architectures and algorithms that have not been adapted as widely due to the difficulty in implementation and reproduction of the results.
>
> As you already commented that our intuitive and efficient approach shows really surprising/novel results, we hope you focus more on the advantage of the method, as it has a **great potential to lower the barrier for video-related research by significantly reducing the computations, memory usage, and training time while preserving the models' performance, which will save time and cost of the researchers as well as reduce the carbon emission.**
>
>
> [1] Srivastava et al., “Dropout: A Simple Way to Prevent Neural Networks from Overfitting”, JMLR 2014
> [2] Lester et al., “The Power of Scale for Parameter-Efficient Prompt Tuning”, EMNLP 2021
> [3] Wei et al., “Chain-of-Thought Prompting Elicits Reasoning in Large Language Models”, NeurIPS 2022
> [4] Houlsby et al., “Parameter-Efficient Transfer Learning for NLP”, ICML 2019
> [5] Hu et al., “LoRA: Low-Rank Adaptation of Large Language Models”, ICLR 2022

---

> ### Author Response · Authors · 2023-11-19
>
> > ***W2-1. 800-epoch pre-training training should be conducted for a fair comparison.***
> $\rightarrow$
> We first respectfully hope that you could understand the limited budget of our research group. We want to emphasize this work is on academic paper to cut down the immense scale of video unsupervised learning, rather than industrial paper using extreme resources.
>
> Additionally, we strongly believe our result on K400 dataset is **clearly a fair comparison** with the entirely identical setting for all methods, and results with more epochs like 200$\rightarrow$800) **does not give us new insight into the tendency of the performance** and we *fairly* validated the strengths of our proposed EVEREST through extensive experiments on multiple architectures (ViT-S/B/L), benchmarks (UCF101, HMDB51, SSv2, K400, Ego4D), and tasks (action recognition, OSCC) with sufficient analyses in our main paper.
>
> Moreover, **we already provided very long training epochs results in our main paper**: We validated our method by training 4800 and 3200 epochs** on the UCF101 and HMDB51 datasets (Please see Table 7), respectively, and achieved superior fine-tuning accuracy along with memory/training time/computational efficiency by a large margin (Please see Table 2 and Figure 5 in the paper).
>
> We sincerely hope the reviewer understands our point that we have already provided ***extensive and fair comparisons*** throughout the paper.
>
> ---
>
> > ***W2-2. & W3. Measuring the wall-clock time memory consumption to show efficiency is not an accurate claim since different devices are used. The ablation study is disorganized since the author uses different datasets and different models, even different devices.***
>
> $\rightarrow$ This is a clear misunderstanding. Throughout Table 3, 4, and 5, we aim to show ours **consistently and stably outperforms baselines on efficiency regardless of hardware and experimental setups**, and we do believe **these multiple evaluations with diverse scenarios make our claim stronger**, rather than weaker.
>
> We confirm that we measured memory/wall-clock time/performance for the comparison in each table with identical experimental setups with the same devices for all baselines and ours. **There is no doubt that our results are accurate and fair**.
>
> However, we expect that an important point of confusion for the reviewer is that memory usage can vary due to differences in hardware. We want to politely emphasize that **it is a very well-known commonsense that different architectures have varying efficiencies in memory utilization, instruction sets, and internal scheduling, caching \& memory management. This means they handle processes differently, which can lead to **different memory usage patterns even for the same model**. As we already clarified, for Tables 3 and 4, the reported result is about the memory size of the **pre-training**. But one difference is that we deploy models on A6000 in Table 3 and A100 in Table 4, where A100 is probably a newer generation than A6000.
> - Therefore, there could be variations in their specific implementations. These differences may impact how memory is utilized or managed, even under similar workloads.
> - Both GPUs have tensor cores optimized for AI workloads, but there might be differences in the number or efficiency of these cores between the two models. This can affect how computations are performed and, consequently, how memory is used.
> - GPUs employ various caching strategies and memory management techniques to optimize performance. Differences in these strategies between the A100 and A6000 could result in different patterns of memory usage.
>
> On the other hand, for Table 5, the reported result is about the memory size of the **fine-tuning**, as clarified in its caption.
>
> We hope you now could understand that the different values in Tables 3,4, and 5 are not our weaknesses but are **strong evidence demonstrating the robustness of our EVEREST across different hardware**.
>
> ---
>
> > ***W2-3. The proposed method only achieved comparable performance under limited budgets, so "state-of-the-art" performance should be carefully used in the abstract.***
>
> $\rightarrow$ Thank you for constructive comment! We understand the reviewer's suggestion and agree with it. We toned down this sentence in the abstract in our latest revision.
>
> *"memory-heavy state-of-the-art methods"* $\rightarrow$ *"memory-heavy baselines"*

---

> ### Author Response · Authors · 2023-11-19
>
> > ***Q1. How do you choose the hyper-parameters for the proposed token selection method?***
>
> $\rightarrow$ As described in experimental settings in Section 5, we follow the same training protocols as VideoMAE. For hyperparameters like $\rho_{pre}$ for our EVEREST, we select them through a standard hyperparameter tuning strategy via the validation set.
>
> ---
>
> > ***Q2. What is the effectiveness of the proposed frame-sampling method?***
>
> $\rightarrow$ We note that our adaptive frame selection helps **reliable and robust video learning** by determining the **most important frames in longer-range video exploration**. This is a unique advantage of our method that cannot be achieved with uniform frame selection, as evidenced by the performance improvements mentioned earlier.
>
> Unedited videos, such as egocentric videos from wearable cameras, may include many **highly redundant or noisy frames**. That is, uniform frame selection with long frame intervals may be more affected by such noisy frames and inconsistencies in the amount of information between incoming frames (Please see**Figures 7 and 10** in our original paper). This may deteriorate the performance of video representation learning methods with uniform sampling as they **cannot distinguish temporally crucial information** in a long video clip and sample frames according to the **pre-defined interval scale**.
>
> Please note that our adaptive frame selection improves the performance of the proposed video representation learning framework by **+0.93%** (+0.68 %p) on OSCC task from Ego4D compared to the uniform frame selection strategy, even for the action recognition task on which the base model without frame selection already achieves high performance (~ 73%), and we also emphasize that this performance gain is statistically significant**.

---

> > ### Comment · Reviewer_4Vgt · 2023-11-21
> >
> > Thanks for pointing it out. This reply addresses this concern.

---

> ### Author Response · Authors · 2023-11-21
> **Dear Reviewer 4Vgt - A Gentle Reminder**
>
> Dear Reviewer 4Vgt,
>
> We are sincerely grateful to you for reading our response.
>
> During the rebuttal period,
>
> - We have clarified multiple **critical differences between the idea of K-centered sampling and ours**.
> - Also, we have clarified the **critical limitations of running-cell masking** used in VideoMAE-V2 and provided a **quantitative** and **qualitative comparison between VideoMAE-V2 (and MAR) and ours** in the revision.
> - We have emphasized that we already provided **very long training epochs** (HMDB51, UCF101), and we really hope you could understand the limited training budgets in our research group. Instead, we verified the efficacy of our proposed method **through extensive empirical evaluations** in terms of architectures, tasks, datasets, and visualizations.
> - We have clarified that **all experiments** in our submission were **clearly fair and reliable**, with identical training/evaluation settings with baselines.
> - Further, we have corrected the reviewer's misunderstanding/confusion of memory consumption from our results - **different architectures** have varying efficiencies in memory utilization, instruction sets, and internal scheduling, caching & memory management. This means they handle processes differently, **leading to different memory usage patterns even for the same model**. Through evaluations using multiple devices, we aim to show that **ours consistently and stably outperforms baselines** on efficiency, **regardless of hardware and experimental setups**.
>
> We remain committed to further improving the quality of our paper by addressing any remaining concerns and suggestions where necessary. With that in mind, if you might have any further feedback, please let us know. We would be grateful for the opportunity to address them and make our work a more solid and valuable contribution to the field of video representation learning.
>
> **Also, we would like to kindly suggest your reconsideration of the rating**, if you feel that our work does not have major concerns with respect to evaluation, resources, reproducibility, and ethical considerations. We understand that the criteria for rating a paper can sometimes be subjective; however, we believe that our work can meet the requirements of the above score, given that most of your concerns are effectively addressed and as long as there are no major issues.
>
> We thank you so much for your time and effort in reviewing our paper, and your constructive feedback that has greatly contributed to improving our paper.
>
> Warm Regards,
> Authors

---

> ### Comment · Reviewer_4Vgt · 2023-11-21
> **Clarify about my understanding about significance.**
>
> Apologies for any confusion. My statement is, "The significance of an experimental-based paper mainly relies on the results or improvement over current state-of-the-art or strong baselines." Thanks for bringing up these works. In [2], the results of "prompt tuning beat GPT-3 prompt design by a large margin, with prompt-tuned T5-Small matching GPT-3XL (over 16 times larger), and prompt-tuned T5-Large beating GPT-3 175B (over 220 times larger)." In [3], "prompting a 540B-parameter language model with just eight chain of thought exemplars achieves state-of-the-art accuracy on the GSM8K benchmark of math word problems, surpassing even fine-tuned GPT-3 with a verifier." My point is that a simple method can be significant, but its significance can be highly relied upon for its superior effectiveness.  My justification regarding this weakness still holds since the performance over the VideoMAE is marginal (sota comparison) under a limited-computation scenario.
>
> The ablation for running cells can be very helpful; thanks for providing it.  But I wonder if you can incorporate more sampling-based methods like K-centered for a more comprehensive ablation and maybe more datasets like SS-V2 to enhance your effectiveness. That would make your statement stronger.

---

> ### Comment · Reviewer_4Vgt · 2023-11-21
> **Training schedule in MAE**
>
> I'm afraid I have to disagree that  "with more epochs like 200 to 800) does not give us new insight into the tendency of the performance".  To the best of my knowledge, I have not seen MAE-based methods without introducing extra supervision have a pre-trained period of 200 epochs, especially for large models.  Since I think the unsupervised pre-training needs a long period to learn useful features.  Their original papers also show a clear performance gap between 200 epochs and 1600 epochs. That is why I think 800 or more epochs is fair for performance comparison.  Could you please give me some references about why you chose such a schedule for comparison? Since I don't know if your method can scale well to more training epochs (larger model) on the large-scale dataset, it also influences my justification regarding your method's significance.

---

> ### Author Response · Authors · 2023-11-22
> **Response to significance**
>
> #### We are sincerely grateful to you for reading our response, and we are glad that **you seem to have solved most of the concerns raised in the initial review**, including comparison with **running-cell masking**, **fairness** of our quantitative analyses, interpretation of tables for **wall-clock time and memory consumption**, **hyperparameter choice**, and the **effectiveness** of the proposed **information-intensive frame sampling**.
>
> #### The following are our responses regarding the remaining concerns, and we hope these address your concerns appropriately.
>
> ---
>
> > **Significance** in terms of **_efficiency_**
>
> #### We politely ask you to consider the significance in terms of **_efficiency_** as well. YOLO [1], which is well known for its efficiency, was less accurate than the SoTA models at the time, but it made a big impact in the community due to its practicality.
>
> #### We would like to emphasize that the **goal** of this work is to make the masked video modeling (MVM) more **_efficient_** _without performance drop_ rather than just achieving state-of-the-art performance.  Several works [2,3,4] reducing redundant tokens of ViT in the image domain have achieved its efficiency but sacrificed performance drop. However, we elaborate on how to effectively reduce redundant tokens in the spatiotemporal domain and find a way to achieve efficiency without performance drop. Therefore, we would like to highlight that it is meaningful to increase efficiency without losing performance. Specifically, our method can save computations up to **45%** and **48%** in pre-training and fine-tuning, respectively, and requires up to **4 × smaller** memory consumption As a result, the computation and memory efficiency of our EVEREST enables us to train MVM models using only **single node** with 8 GPUs, which helps the vision community to **lower the barrier to further SSL for video research.**
> #### Moreover, unlike other methods, our method is applicable not only in pre-training (PT) but also in fine-tuning (FT), which can **further** save more computation and memory requirements for the PT-FT process in total.
> ####  Therefore, to the best of our knowledge, our method is the **_first_** work to enhance the efficiency for both PT and FT in the video-ssl literature.
>
>
> ---
> #### [1] Redmon et al. You Only Look Once: Unified, Real-Time Object Detection. CVPR 2016.
> #### [2] Rao et al. Dynamicvit: Efficient vision transformers with dynamic token sparsification. NeurIPS 2021.
> #### [3] Liang et al. Expediting vision transformers via token reorganizations. ICLR 2022.
> #### [4] Bolya et al. Token merging: Your ViT but faster. ICLR 2023.

---

> ### Author Response · Authors · 2023-11-22
> **Response to comparison with K-centered [1]**
>
> > **Comparison with K-centered [1]**
>
>
> We agreed that we could support this clear proposition, *"selecting the farthest tokens based on the algorithm [1] doesn't mean minimizing redundancy"*, through an ablation study.
>
> But we sincerely hope you could understand the time constraints that we only left a **single day** for discussion, which is insufficient to provide additional analyses that the reviewer suggested. However, we definitely will add the suggested ablation study on K-centered using ViT backbones and evaluation on SSv2 with in-depth discussions in our final revision.
>
> One thing we note is that **K-centered [1]** basically shows **degenerated** (or on-par) performance on **ViT** backbone (*Table 2 in the K-centered paper*) compared to **TimeSformer** backbone (which we compared in our paper), showing 2.33\% lower top-1 accuracy (*77.98 v.s. 75.65*).
>
> ---
>
> Regarding the *fairness of the comparison*, we **kindly disagree with the claim that our comparison lacks fairness**.
>
> **Most recent papers on video understanding directly compare** their method with baselines using different backbones since **many works can be backbone-tailored or tuned under specific backbone structures**, and it is infeasible to validate the significance of methodology through many diverse backbones as video understanding works are basically **very heavy and computationally-/financially-expensive**. (Here, we want to **emphasize** the contribution of our **EVEREST** again, which achieves **surprising efficiency and practicability** while showing competitive performance against dense & strong models. This is a **very significant and practical contribution** to the video understanding community.)
>
> That is why recent works **provide GFLOPs, Memory (or parameters), and wall-clock training time** for a **fair comparison across prior works using multiple different backbones**. This evaluation paradigm is also clearly observed in the main tables (*Tables 5 and 6*) of **VideoMAE paper [2]**. We don't believe there's any doubt that the many previous studies on video understanding, including VideoMAE, have made a fair comparison.
>
> In the end, **we strongly believe it is very clear** that **our EVEREST achieves superior performance over K-centered paper**. On the other hand, as said, we will definitely **provide a comprehensive and extensive analysis/ablation study** comparing with K-centered [1] in our final revision. We strongly agree that this potential analysis further strengthens our submission.
>
> [1]  Park, Seong Hyeon, et al. "K-centered patch sampling for efficient video recognition." ECCV 2022
> [2] Zhan Tong, et al. "Videomae: Masked autoencoders are data-efficient learners for self-supervised video pre-training, NeurIPS 2022

---

> ### Author Response · Authors · 2023-11-22
> **Response to longer training schedule**
>
> #### The reason we chose 200 epochs is that **with the limited computing resource (at the academic level)**, we wanted to focus the **generalization** of our method across multiple models (ViT-S/B/L) and multiple benchmarks (K400/SSv2/UCF101/HMDB/Ego4D), rather than achieving SOTA only on K400 and SSv2 datasets for longer training. We also validated our method by training 4800 and 3200 epochs on the UCF101 and HMDB51 datasets (Please see Table 7), respectively. From the results of various benchmarks with different backbone models, we can expect our model to show consistent performance gains over longer training. Therefore, we respectfully disagree that simply not having enough pre-training epochs in a particular dataset seriously flaws the contribution of our method.
>
> #### For another excuse,  estimated training GPU Cloud [1] costs for VideoMAE and our EVEREST for 800 epochs using 8 A6000 GPUs are as follows:
>
> |Models w/ ViT-B|pre-training time (h) |cost ($)|
> |----------|----------------|------|
> |VideoMAE | 249.3 | 2,000 |
> |Ours | 110.6 | 888|
>
> #### To training all models over 800 epochs on K400 and SSv2, it requires vast training cost. So we had no choice but to train for 200 epochs and rather distributed the cost of validating more models on diverse benchmarks.
>
> we hope you understand our limited situation at this academic level.
>
> ---
>
> #### [1] https://lambdalabs.com/blog/introducing-nvidia-rtx-a6000-gpu-instances-on-lambda-cloud

---

> ### Author Response · Authors · 2023-11-23
> **Discussion period ends within 8 hours**
>
> Dear Reviewer 4Vgt,
>
> We really appreciate your effort in reviewing our submission, and glad that you seem to have solved most of the concerns raised in the initial review.
>
> Since the discussion period for ICLR 2024 ends within 8 hours, we politely ask you to read our new responses by any chance. Please understand that we have made our best effort to address your concerns during this period.
>
> Also, we would like to kindly suggest **a reconsideration of the initial rating (reject: 3)**, if you agree that **many concerns/misunderstandings raised by the initial review are resolved**. We strongly believe that most of your concerns are effectively addressed as long as there are **no significant issues to be a score of 3**.
>
> We thank you so much for your time and effort in reviewing our paper and for the constructive feedback that has greatly improved it.
>
> Best,
> Authors

---

> > ### Comment · Reviewer_4Vgt · 2023-11-23
> >
> > Thank you for reminding me. I will maintain my current rate for this period as my primary concern has not yet been addressed. The effectiveness of your work mainly depends on training efficiency, but the major issue lies in the limitations of computation for downstream results. In your scenario, the greatest challenge lies not in training efficiency, but in performance. It would be truly impressive if the model could be significantly trained and improved under 200 epochs, utilizing eight GPUs. Since 200 epochs are not computationally intensive, it would be helpful if you could provide references from the literature where this setting has been used. This way, you can compare your results with theirs to demonstrate the superiority of your performance. However, I have not seen any remarkable improvement in performance thus far. Therefore, I request a longer training schedule to demonstrate that your method can indeed achieve state-of-the-art (SOTA) performance, while also being at least 2X faster. Such results would be highly impressive, but currently, I haven't seen evidence of both.

---

> ### Author Response · Authors · 2023-11-23
> **Please do not confuse that achieving overwhelming accuracy against SoTA models is not our interest**
>
> Thanks for your response.
>
> We want to emphasize again that achieving superior accuracy compared to recent SoTA video representation learning is not our primary interest at all. **We sincerely ask the reviewer not to confuse our major contributions.**
>
> Our primary focus and significant contribution is **maximizing the efficiency in terms of memory, computation, and financial efficiency** during pre-training and fine-tuning while maintaining competitive performance compared to strong \& heavy baselines. This can be clearly represented throughout our paper and title - ***Efficient** Masked Video Autoencoder by **Removing Redundant** Spatiotemporal Tokens*, as well as we have repeatedly clarified this in our rebuttals multiple times.
>
> We have already shown around **$4\times$ memory efficiency, $2\times$ computational efficiency, $2\times$ training time efficiency, and $3\times$ financial efficiency** while achieving **competitive performance compared to recent memory-heavy baselines**. We strongly believe these contributions are sufficiently significant to be accepted in the venue.
>
> Additionally, we want to politely refer to the **_Reviewer guidelines_** of the ICLR [1]: ***What is the significance of the work?** Does it contribute new knowledge and sufficient value to the community? Note, **this does not necessarily require state-of-the-art results**. Submissions bring value to the ICLR community when they convincingly demonstrate **new, relevant, impactful knowledge***.
>
> We do not believe that the lack of assurance of achieving SoTA accuracy can be the reason for a score of 3 and respectfully suggest a reconsideration of the initial rating.
>
> Best,
> Authors
>
> [1] https://iclr.cc/Conferences/2023/ReviewerGuide

---

### Author Response · Authors · 2023-11-19
**General Comment by Authors**

We would like to thank the reviewers for their constructive comments and suggestions. We appreciate the reviewers' comments that the paper is **well-written and clear** (Reviewer JSFH, viqa), **simple and straightforward** (Reviewer 4Vgt, viqa), and **extensively evaluated** (Reviewer 4Vgt, viqa). We further thank Reviewer JSFH for commenting that our method is **novel and would be of interest to the ICLR community**.


During the rebuttal period, we have made every effort to address all the reviewers' concerns faithfully. We clearly answered all questions from the reviewers and provided additional experimental results and analyses that the reviewer requested. We believe that these responses effectively resolve the reviewer's concerns. And we additionally compared our method with running-cell masking [3], which is used in VideoMAE-V2 [2]. **We provide clear limitations in running-cell masking that VideoMAE-V2 suffers from visual redundancy and cannot capture semantic importance (Please see Figure 8 in our latest revision).**

For quantitative analysis, during the rebuttal period, we have compared the effect of our redundancy-robust token masking with running-cell masking used in MAR [3] and VideoMAE-V2. We pre-trained with these masking strategies and performed conventional full-finetuning. As shown in the table below, our** EVEREST clearly outperforms running-cell masking**, demonstrating our effectiveness against other selection strategies given the same architecture.

| method |UCF101|HMDB51|
|---------------|------------|-----------|
| Running cell | 91.0 | 61.4 |
| EVEREST (Ours)| **93.4** | **68.1**|


[2] Wang, Limin, et al. "Videomae v2: Scaling video masked autoencoders with dual masking." CVPR 2023
[3] Qing et al., MAR: Masked Autoencoders for Efficient Action Recognition, IEEE Transactions on Multimedia, 2023

Many researchers/students in academia have struggled with exhaustively large memory, training time, and computational costs with only a small number of GPUs (e.g., single-node equipped with 8 GPUs) because self-supervised learning for video data [1,2,3] requires immense computations and memory resources. In particular, VideoMAE [1] uses **8 nodes with 64× V100 GPUs**, and ST-MAE [2] uses **16 nodes with 128× A100 GPUs**.

However, our EVEREST pruning redundant visual tokens can reduce computation burdens during both pre-training and fine-tuning, saving computations up to **45%** and **48%** in pre-training and fine-tuning, respectively, and requiring up to **4 × smaller memory consumption** without performance drop (please refer to Figure 1 in the original paper). The memory-efficient training results show our method is much more efficient and practical than the previously proposed random-based [1,2,3] and similarity-based [4] approaches.

As a result, **the computation and memory efficiency of our EVEREST** enables us to train MVM models using only a single node with 8 GPUs, which **helps the vision community to lower the barrier to further SSL for video research**.


[1] VideoMAE: Masked Autoencoders are Data-Efficient Learners for Self-Supervised Video Pre-Training. NeurIPS 2022.
[2] Masked autoencoders as spatiotemporal learners. NeurIPS 2022.
[3] Videomae v2: Scaling video masked autoencoders with dual masking. CVPR 2023.
[4] K-centered patch sampling for efficient video recognition. ECCV 2022.

---

### Meta-Review · Area_Chair_VLx7 · 2023-12-02

**Metareview:**

The submission addresses an important problem – efficient modeling for video understanding – and aims to exploit the inherent redundancy in videos. The paper presents appreciably intuitive and effective techniques for token selection and optimizations for salient frame selection. It achieves approximately a 50% reduction in memory and compute on VideoMAE, while maintaining top-1 performance on several benchmarks under the same settings. This contribution is well appreciated by all reviewers. However, the reviewers have pointed out several weaknesses. The most prominent is: 1) a lack of adherence to common practice in comparison experiment design. Mainly, this involves using 200 epochs rather than the 800 epochs employed in K400 pre-training. The authors justify this with a lack of sufficient compute budget. The reviewer's point is that this work would be more impressive if it were trained longer and achieved the same 2x efficiency while attaining state-of-the-art numbers. However, they did show long fine-tuning on UCF101 and HMDB5, with favorable outcomes. 2)Missing comparison to other baselines (AdaMAE, MME, MVD, St-MAE, and Omnimae), and newer baselines after VideoMAE (a NeurIPS’22 paper). In the rebuttal, the authors provided new results that support the proposed technique by comparing it to VideoMAE-v2, a newer CVPR’23 baseline. Some of the requested baselines have also been included in the paper, but not all. However, these have not abated the concerns of two reviewers.

I would also like to stress that the authors have made a great effort in engaging the reviewers, providing detailed responses, as well as presenting additional experimental results that addresses certain concerns.  However, I am not an expert on the specific domain to overwrite reviewers' concerns.

**Justification For Why Not Higher Score:**

The paper received 1x reject, 1x below acceptance threshold and 1x accept. The need for more comprehensive comparisons, and experiment design in pre-training are the main concerns. The computational savings are significant. However, questions remain if these gains will persist when trained at typical long training regimes, while maintaining SOTA in such settings.

**Justification For Why Not Lower Score:**

N/A

---

### Decision · Program_Chairs · 2024-01-16

Reject